# ROBULT: A SCALABLE FRAMEWORK FOR SEMI-SUPERVISED MULTIMODAL LEARNING WITH MISSING MODALITIES

## ABSTRACT

In multimodal learning, the presence of missing modalities and limited labeled data presents significant challenges for building robust models. We propose **Robult**, a novel framework designed to address these challenges by leveraging an information-theoretic approach to preserve modality-specific features and synergistic information across modalities. Our model introduces two key objectives: (1) a latent reconstruction loss to retain unique modality-specific information, and (2) a novel soft Positive-Unlabeled (PU) contrastive loss to efficiently utilize sparse labeled data in semi-supervised settings. Robult seamlessly integrates into deep learning architectures, enhancing performance across multiple downstream tasks and ensuring robustness even when modalities are missing at inference time. Empirical results across diverse datasets demonstrate that Robult surpasses existing methods in handling both semi-supervised learning and missing modalities, while its lightweight design enables scalability and easy integration with existing frameworks.

## 1 INTRODUCTION

**Motivation:** In the Big Data era, the continuous volume of data from diverse sources and formats necessitates robust multimodal processing while minimizing the need for extensive manual labeling. Multimodal learning is a method to process multiple data sources in parallel and outperforms traditional machine learning on a single modality, e.g. Huang et al. (2021), demonstrating its potential across various applications such as language-vision interaction Li et al. (2022); Talmor et al. (2021), machine translation Yao & Wan (2020), communication Lazaridou et al. (2020), healthcare Soenksen et al. (2022); Chen et al. (2021a), robotics Miralles et al. (2022), and finance Windsor & Cao (2022). However, most existing methods such as Daunhawer et al. (2023); Peng et al. (2022); Chuang et al. (2022); Ma et al. (2022); Trosten et al. (2021); Hadji et al. (2021) consider ideal scenarios where the data are fully labeled and all modalities are present. Our research addresses these two real-world challenges for multimodal learning: (1) semi-supervised scenarios with missing labels during training, and (2) occlusion or corruption scenarios where some modal data are missing during inference.

The semi-supervised learning setting, arises from the practical challenges associated with labeling raw data, particularly in domains where explicit labels are not readily available or are labor-intensive to acquire. This problem is exacerbated with multimodal learning tasks where each modality needs to be labeled individually, such as object segmentation of video and lidar data in autonomous driving Zhang et al. (2022); Caesar et al. (2020), or medical segmentation of different imaging modalities which requires diverse expert knowledge and often lacks a standardized labeling procedure Acosta et al. (2022). While there have been modality-specific advancements in semi-supervised learning using techniques like knowledge distillation Su et al. (2021) and pseudo-labeling Aberdam et al. (2022), these methods are struggle to generalize to multi-modality scenarios, particularly when some modalities

Along with missing training labels, a prevalent challenge in multimodal model deployment (i.e. in inference time) is the corruption or missing of some modal data. For example, an autonomous car with a camera obscured by mud may rely solely on lidar data, or a medical diagnosis system might only access one imaging modality in resource-limited hospitals. Research on this issue has explored generative strategies, such as VAE variations Wu & Goodman (2018) to reconstruct missing

modalities, and transfer learning to align latent spaces for cross-modal knowledge transfer Ma et al. (2022); Lee & Van der Schaar (2021); Wang et al. (2020). Recent generative methods Feichtenhofer et al. (2022); Woo et al. (2023) require specialized architectures and lack flexibility, while transfer learning approaches, though more adaptable, often depend on intuition and complete labels during training Chen et al. (2023). Existing methods using these approaches have not addressed the issue of missing labels during training.

Thus, the joint consideration of semi-supervised training with missing modality inference presents an unresolved challenge. Semi-supervised methods Lian et al. (2022); Zheng et al. (2022) often assume the availability of all modalities, making their pipelines dysfunctional when inputs are incomplete. Conversely, missing modality methods avoid the challenge of scarce labels in the training data Woo et al. (2023). Nonetheless, a model deployed in real-world applications (e.g. autonomous vehicles or medical centers) should leverage unlabeled data during training and exhibit robustness to missing modalities during inference. We present a multimodal framework explicitly designed to accomplish this through a novel soft Positive-Unlabeled (PU) contrastive loss and modality-specific reconstruction losses. Our method, backed by information theoretic bounds, performs well on diverse datasets demonstrating that it can generalize across different modalities and tasks.

**Our approach:** Our model maintains its representation capacity under scarce training labels and missing modalities by maximizing the mutual information between the model's learned representations of the input and the target task output. Inspired by Partial Information Decomposition Williams & Beer (2010), we note that the mutual information provided by an input $X$ with $M$ modalities $(X^1, \ldots, X^M)$ for a given task $Y$ can be broken down into the following quantities:

$$\mathcal{MI}(\{X^1, \ldots, X^M\}; Y) = \mathcal{R}(\{X^1, \ldots, X^M\}; Y) + \sum_{i=1}^{M} \mathcal{U}(X^i; Y) + \mathcal{S}(\{X^1, \ldots, X^M\}; Y),$$
(1)

in which $\mathcal{R}(.;.)$ quantifies redundancy - the task-related shared information of $M$ input variables; $\mathcal{U}^i(.;.)$ represents the unique information of the $i^{th}$ modality; and $\mathcal{S}(.;.)$ denotes synergy information, which is the knowledge generated by the interaction among $M$ modalities.

From an information-theoretic perspective, existing knowledge distillation Chen et al. (2023) and contrastive learning Radford et al. (2021) approaches implicitly reproduce $\mathcal{S}(.;.)$ and $\mathcal{R}(.;.)$ through intuitive alignment processes in a latent space. In contrast, in this work we explicitly model these values and introduce a mutual information maximization objective (Objective 2.1) as a loss term. We establish a theoretical lower bound for this target quantity, which can be maximized using our PU contrastive loss. This innovative loss uses a novel soft-positive pseudo-labeling scheme for unlabeled data. By accounting for the uncertainty of pseudo-labels, our approach distinguishes itself from existing semi-supervised methods.

Furthermore, we observe that latent-space alignment methods Chen et al. (2023); Radford et al. (2021) diminish the distinct information given by each modality to the representation, $U(.;.)$. We hypothesize that this modality-specific information is beneficial to the overall performance when label information is scarce and other modalities may be missing. Therefore, we frame the retention of each modality's unique information $\mathcal{U}^i(.;.)$ as Objective 2.2 and achieve its upper bound through a simple reconstruction procedure in the latent space. This procedure is universally applicable across different data modalities, and indeed improves performance as demonstrated by our results and ablations (Tables 1 and 3).

Together, Objectives 2.1 and 2.2 form a semi-supervised multimodal learning method that is robust to missing modalities. We refer to this proposed method as Robult, the Robust Multimodal Pipeline, hereafter. The superior performance of Robult compared to existing methods in comprehensive empirical experiments (Section 3), as well as Robult's theoretical underpinnings, underscores its promise toward various real-world settings. Our main contributions can be summarized as follows:

- We jointly address two common real-world problems: semi-supervised training and missing modalities during evaluation. This generalized setting yields a distinct challenge for existing works, yet Robult has proven effective through rigorous experimentation.

- We frame two objectives under an information-theoretic viewpoint and derive suitable loss strategies to attain these goals.

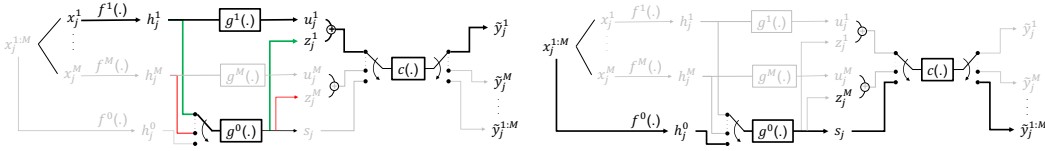

(a) Inference when some modalities are missing          (b) Inference when all modalities are available

Figure 1: In the Robult Pipeline, for an input $x_j^{1:M}$ (1a) $f^i$ and $g^i$, for $i = 1 \ldots M$, extract latent and unique-info features, $h_j^i$ and $u_j^i$, and use a shared module, $c$, to predict outputs $\tilde{y}_j^i$. (1b) $f^0$ and $g^0$ use all the modalities jointly to extract a fused latent vector and synergy, $h_j^0$ and $s_j$, which is used in $c$ with final output $\tilde{y}_j^{1:M}$. During training the synergy module $g^0$ also produces modality-specific $z_j^i$, used in loss calculations to encourage the unimodal branches to approach multi-modal performance.

- We introduce a novel soft Positive-Unlabeled contrastive loss that efficiently utilizes limited labeled information through selective weighting of potential positives.

## 2  METHODOLOGY

**Method Overview:** To jointly address the lack of labels during training and the absence of modalities during evaluation, we introduce a versatile pipeline named Robult (Figure 1). Robult consists of $M$ modality-specific branches and a fusion branch indexed with zero. This system enables each uni-modal branch to learn and replicate the synergistic information from the fusion branch with minimal supervision, thereby closing the performance gap between the scenario where only one modality is available and the scenario where all modalities are present.

All branches are executed during training, and three loss functions update the learned modules. The synergy-based soft positive unlabeled loss, $\mathcal{L}_{PU}$, maximizes knowledge extraction from the few labeled samples in a batch (Subsection 2.1). The reconstruction loss, $\mathcal{L}_{rec}$ forces each branch to extract unique modality-specific information (Subsection 2.2). The task-specific supervised loss, $\mathcal{L}_{sup}$ is used on the labeled samples across all modules to learn label information (Subsection 2.3). During testing, if all modalities are present, Robult uses the fusion branch, shown in Fig. 1b. Otherwise, if one or some modalities are missing (Fig. 1a), Robult uses the branches corresponding to available modalities, then performs decision-level fusion (e.g. simple averaging) (discussed more in Appendix C.1, Table 6).

**Notation:** In this study, we consider the scenario where samples in the training dataset contain all modalities, but an arbitrary number of the training samples are unlabeled (Figure 2). Thus, our training setup covers various labeled:unlabeled ratios, ranging from a supervised setting (all data is labeled) to an unsupervised setting (no data is labeled). Let a training dataset with $n$ samples be $\mathcal{X} = \{x_1, \ldots, x_n\}$, where the $j^{th}$ data point, $x_j = (x_j^1, \ldots, x_j^M)$, contains $M$ modalities. Assuming the first $k$ samples of $\mathcal{X}$ are labeled, the corresponding label set $\mathcal{Y} = \{y_1, \ldots, y_k\}$ consists of $k$ samples, where $k < n$.

Let $f^i$, $g^i$, and $c$ indicate learned modules where $i$ corresponds to the $i^{th}$ modality. Their inputs, $h_j^i$, $u_j^i$, and $z_j^i$, indicate each modality's latent, unique-information, and synergy-based representations respectively, with $s_j$ being the joint synergy across all modalities. For brevity, we use index 0 to denote the latent variables generated using all the modalities jointly $x_j^{1:M}$, i.e. $h_j^0$ is the joint latent representation of all the modalities of the $j^{th}$ data point generated by the fusion network $f^0(x_j^1, x_j^2, \ldots x_j^M)$. Throughout our theoretical analysis, we primarily use the notation $X^i$ (or $H^i, U^i, Z^i$) to represent the random variables associated with the $i^{th}$ modality input, and $S$ for the synergy random variable from all modalities.

Instead of operating directly on a data point $x_j$, we first project raw inputs into $m$ latent vectors, one for each modality, i.e. $f^i(x_j^i) = h_j^i$ for $i = 1, \ldots, M$. The benefits of working with the latent feature vectors are twofold: (1) it ensures that Robult can be generalized to modalities with different preferred projection methods, (2) various fusion strategies can be adopted to efficiently attain fused representations as fusing raw data presents more challenges.

**Semi-supervision:** To address the setting of label scarcity, we aim to leverage the similarity of the synergy, i.e. the fused representation $H^0$, to the unimodal representations, $H^i$, in order for the model to learn from unlabeled data samples in a batch. However, the $H^i$'s and $H^0$ cannot be directly compared as they were projected into different latent spaces by their respective $f^i(.)$'s. Thus we employ the synergy module $g^0(.)$ first on the joint representation of all the modalities, $g^0(H^0) = S$, and then on each modality's representation $g^0(H^i) = Z^i$ for $i = 1, \ldots, M$ (Fig.1). Finally, we apply a novel achievable contrastive loss, termed Soft Positive-Unlabeled (PU) Contrastive Loss, on the $S$ and $Z^i$ terms as detailed in Section 2.1.

Theoretically, this PU contrastive loss corresponds to a mutual information maximization problem between the desired fused representation $S$ and the learned unimodal representation projected into that same latent space $Z^i$'s. Thus Objective 2.1 of our method is expressed as follow:

**Objective 2.1.** Aligning $S$ and $Z^i$ by maximizing the mutual information $\mathcal{MI}(S, Z^i)$   $(i = \overline{1, M})$.

**Missing modalities:** Next we consider the setting where labels may be scarce during training in addition to potentially missing modalities during testing. Common approaches to dealing with missing modalities are transfer learning via knowledge distillation or contrastive loss Poklukar et al. (2022b); Garcia et al. (2021); Stroud et al. (2020). These methods often rely on the synergy/alignment between modalities to retrieve labels during inference when one modality is missing. However, when training labels are sparse, the alignment between modalities and labels is weaker and these methods fail. We observe that attempting to only align modalities during training diminishes the unique information provided by each modality, thus the model is losing information that could help inform it of the label during inference. Therefore, in the circumstance where labels are missing during training *and* modalities are missing during testing, a model would benefit by maintaining the synergy while also explicitly preserving each modality's unique information. This claim is later supported by experimental results and ablations (Section 3 - Table 1).

To address the challenge of vanishing modality-specific information from multimodal alignment during training, we emphasize a disentanglement strategy that preserves unique information while still facilitating the synergy learning process of Objective 2.1. Robult integrates a set of modules $g^i(H^i)$, where $i = 1, \ldots, M$, to produce unique representations $U^i$ for each modality. We aim to preserve the unique information for each modality via the learning of $U^i$ with Objective 2.2, detailed in Section 2.2 and stated as follows:

**Objective 2.2.** Learning $U^i$ by minimizing the conditional entropy $\mathcal{H}(H^i|Z^i, U^i)$   $(i = \overline{1, M})$.

## 2.1 MAXIMIZING MUTUAL INFORMATION WITH SOFT POSITIVE-UNLABELED CONTRASTIVE LEARNING

To mitigate the effect of missing modalities, we should foster alignment between the synergy latent variable $S$ and the unimodal variable $Z^i$, denoted by Objective 2.1, by maximizing their mutual information. However, direct calculation of this quantity is not feasible without knowledge of the joint distribution $p_{S,Z^i}$ or the marginal distributions $p_S$ and $p_{Z^i}$. Therefore, we derive its achievable lower bound and strive to maximize this quantity instead. Define $F$ as a variable that indicates whether a pair $(s_j, z_j^i)$ is drawn from the joint distribution $p_{S,Z^i}$ (where $F = 1$ indicates dependence) or from the product of the marginal distributions $p_S \otimes p_{Z^i}$ (where $F = 0$ signifies independence). The result can be stated as follows.

*Result.* A lower bound of $\mathcal{MI}(S, Z^i)$:

$$\mathcal{MI}(S, Z^i) \geq -\mathbb{E}_{p_{S,Z^i}} \log v(S, Z^i) = -\mathbb{E}_{p(S,Z^i|F=1)} \log v(S, Z^i) \qquad (2)$$

where $v(S, Z^i)$ is a non-parametric approximation of $p(F = 1|S, Z^i)$. Given a couplet $(s_j, z_j^i)$ in a batch of B samples, where $s_j$ is the joint synergy across all modalities and $z_j^i$ is the modality-specific representation for sample $j$ and modality $i$, $v(s_j, z_j^i)$ is given by:

$$v(s_j, z_j^i) = \frac{\phi(s_j, z_j^i)}{\sum_k^B \phi(s_j, z_k^i)}; \quad \text{where} \quad \phi(s_j, z_j^i) = exp(< s_j; z_j^i > /\tau).$$

Detailed derivation of Result 2 is covered in Appendix A.2. In this result, the lower bound still depends on the expectation operator $\mathbb{E}_{p_{S,Z^i}}$, a key source of deviation in existing studies. Commonly, two approaches are used for sampling from the joint distribution $p_{S,Z^i}$: (1) Instance-level sampling, considering pairs $(s_j, z_j^i)$ from different samples $j$ of the same batch Radford et al.

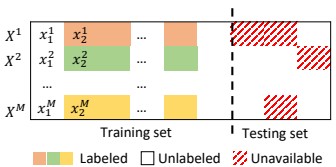

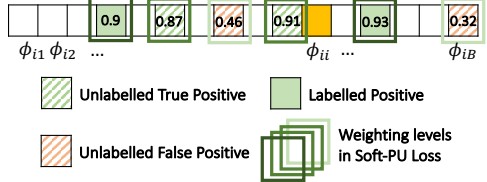

Figure 2: Illustration for training/testing datasets under investigation. The training dataset contains full-modality samples, both labeled and unlabeled.

Figure 3: Soft-PU Loss mechanism. Unlabeled positive pairs are identified using soft labels from Robult's classifier. These pairs are re-weighted based on their proximity and the mean proximity of true labeled positive pairs to mitigate false positives.

(2021); (2) Label-level sampling, involving label information and sampling pairs with the same labels: $(s_j, z_k^i)|y_j = y_k$ Chen et al. (2021b). While performing contrastive learning the former approach is likely to introduce undesired false negative couplets, the latter direction requires fully labeled training data, making either solution less than ideal.

To bridge this label-related gap, we deploy a novel soft Positive-Unlabelled (PU) constrastive loss, together with an adaptive weighting strategy. Let $L$ be a variable used to determine if a pair $< s_j, z_k^i >$ is labeled ($L = 1$) or not ($L = 0$) (where $L = 1$ if the label information of both samples $i$ and $k$ is known, and $L = 0$ otherwise), the lower bound in *Result* 2 can be rewritten as:

$$
\begin{aligned}
-\mathbb{E}_{P_{S,Z^i}} \log v(S, Z^i) &= -\mathbb{E}_{P(S,Z^i|F=1)} \log v(S, Z^i) \\
&= -\mathbb{E}_{P(S,Z^i|F=1,L=1)} \log v(S, Z^i) * p(L=1) \\
&\quad -\mathbb{E}_{P(S,Z^i|F=1,L=0)} \log v(S, Z^i) * p(L=0).
\end{aligned}
\tag{3}
$$

From this step, we formulate the two terms of Eq.3 as two separate lost terms, $\mathcal{L}_{lb}$ for the labeled data and $\mathcal{L}_{ulb}$ for the unlabeled data. Let $B_{F=1,L=1}$ be the index set of the inputs in a batch that have the same true class as anchor element $j$ and are labeled, i.e. $k \in B_{F=1,L=1} \iff (s_j, z_k^i) \sim p(S, Z^i|F=1, L=1)$.

The first term of Eq. 3 can be straightforwardly modeled as a NT-Xent-like contrastive loss Chen et al. (2020):

$$
\mathcal{L}_{lb} = -\frac{1}{M} \sum_{i=1}^{M} \mathcal{L}_{lb}^i; \quad \text{where} \quad \mathcal{L}_{lb}^i = -\frac{1}{||B_{F=1,L=1}||} \sum_{k \in B_{F=1,L=1}} \log v(s_j, z_k^i).
\tag{4}
$$

Regarding the second term, there is no direct solution for sampling from $p(S, Z^i|F=1, L=0)$. To address this, we propose utilizing the output of the Robult classifier as soft label information, which is then regularized by a set of adaptive weights generated within each mini-batch. In the initial training stages, the Robult classifier may exhibit instability, which can lead to unreliable outcomes. This instability is particularly problematic as it can impede the effective filtering of false positives during the sampling process, as illustrated in Figure 3. To mitigate this and enhance the performance of the final loss function, we adjust the contribution of soft-labeled pairs, thereby distinguishing our approach from traditional pseudo-labeling methods Aberdam et al. (2022).

For each anchor sample $s_j$ within a given mini-batch, there are inevitably some labeled positive partners, or at least unimodal representations $z_j^i$ in unsupervised scenarios. The average proximity of these labeled partners to $s_j$ provides a reference for determining what proximity should be considered "positive" for this anchor. The proximity of unlabeled positive partners, as determined by the Robult classifier, should ideally be close to this reference mean. We implement a strategy where the contribution of a couplet to the loss is increased if its proximity closely matches that of the reference couplets and reduced otherwise. This method helps to effectively lower the influence of potential false positives, as demonstrated in Figure 3. The weighting of these couplets is calculated using the RBF kernel, allowing for precise adjustment based on proximity.

$$
w_{jk}^i = RBF(\phi_j^i, \phi(s_j, z_k^i)); \quad \text{where} \quad \phi_j^i = mean\left\{\phi(s_j, z_k^i)|\tilde{k} \sim p(\tilde{k}|j, F=1, L=1)\right\}.
\tag{5}
$$

Let $B_{F=1,L=0}$ be the index set of the inputs in a batch that have the same true class as anchor element $j$ and are not labeled, i.e. $k \in B_{F=1,L=0} \iff (s_j, z_k^i) \sim p(S, Z^i | F = 1, L = 0)$. The unlabled loss term and complete soft Positive-Unlabeled (PU) loss are given by:

$$\mathcal{L}_{ulb} = -\frac{1}{M}\sum_{i=1}^{M}\mathcal{L}_{ulb}^i; \quad \text{where} \quad \mathcal{L}_{ulb}^i = -\frac{1}{||B_{F=1,L=0}||}\sum_{k \in B_{F=1,L=0}} w_{jk}^i \log v(s_j, z_k^i). \quad (6)$$

$$\mathcal{L}_{PU} = \mathcal{L}_{ulb} + \mathcal{L}_{lb} \quad (7)$$

## 2.2 MINIMIZING CONDITIONAL ENTROPY WITH LATENT RECONSTRUCTION ERROR

This section outlines the procedure to achieve Objective 2.2, effectively preserving unique information $U^i$. Let $p_{U^i, Z^i}$ denote the joint distribution of $U^i$ and $Z^i$, where $(u_j^i, z_j^i) \sim p_{U^i, Z^i}$ is generated from the corresponding instance $h_j^i$. Subsequently, we derive $\mathcal{H}(H^i | Z^i, U^i)$ as follows:

$$\mathcal{H}(H^i | Z^i, U^i) = -\mathbb{E}_{U^i, Z^i \sim p_{U^i, Z^i}}\Big[\mathbb{E}_{H^i \sim p(H^i | U^i, Z^i)}\big[\log p\left(H^i \mid U^i, Z^i\right)\big]\Big] \quad (8)$$

Quantifying $p(H^i \mid U^i, Z^i)$ in Eq. 8 is not straightforward, we approximate it by introducing a distribution $q(H^i \mid U^i, Z^i)$:

$$\begin{aligned}
\mathcal{H}(H^i | Z^i, U^i) &= -\mathbb{E}_{U^i, Z^i \sim p_{U^i, Z^i}}\Big[\mathbb{E}_{H^i \sim p(H^i | U^i, Z^i)}\big[\log q\left(H^i \mid U^i, Z^i\right)\frac{p\left(H^i \mid U^i, Z^i\right)}{q\left(H^i \mid U^i, Z^i\right)}\big]\Big] \\
&= -\mathbb{E}_{U^i, Z^i \sim p_{U^i, Z^i}}\Big[\mathbb{E}_{H^i \sim p(H^i | U^i, Z^i)}\big[\log q\left(H^i \mid U^i, Z^i\right)\big] + d_{KL}(p||q)\Big] \quad (9) \\
&\leq -\mathbb{E}_{U^i, Z^i \sim p_{U^i, Z^i}}\Big[\mathbb{E}_{H^i \sim p(H^i | U^i, Z^i)}\big[\log q\left(H^i \mid U^i, Z^i\right)\big]\Big].
\end{aligned}$$

Thus, minimizing $\mathcal{H}(H^i | Z^i, U^i)$ can be relaxed to minimizing its ELBO-alike Kingma & Welling (2013) upperbound with the newly defined distribution $q\left(H^i \mid U^i, Z^i\right)$. In this study, we model $q$ by incorporating a straightforward yet efficient reconstruction procedure in the shared latent space, which involves the module $r^i(U^i, Z^i) = \tilde{H}^i$ and a reconstruction loss. Given a couplet $(u_j^i, z_j^i) \sim p_{U^i, Z^i}$ generated from $h_j^i$, module $r^i(.)$ attempt to recreate $\tilde{h}_j^i$ resemble $h_j^i$ by enforcing latent reconstruction loss:

$$\mathcal{L}_{rec} = \frac{1}{MB}\sum_{i=1}^{M}\sum_{j=1}^{B}1- < \tilde{h}_j^i, h_j^i >^2, \quad (10)$$

in which $<;>$ denotes the $L2$-normalized dot product operation, $B$ is the size of mini-batch, and $M$ is the number of modalities. By performing reconstruction in latent space, this procedure efficiently alleviates the time and computational burden, while streamlining the process of reconstructing various raw modalities; thereby preserving the generality of Robult. We perform back-propagation with $\mathcal{L}_{rec}$ exclusively on the $M$ unimodal branches. This essentially concentrates the impact of this criterion on retaining unique information $\mathcal{U}(X^i, Y)$ through $U^i$, distinct from fostering the shared branch to learn synergy information - discussed in Section 2.1.

## 2.3 TRAINING STRATEGY

The objectives in Sections 2.1 and 2.2 are effectively attained through the discussed loss functions. Given their distinct nature, learning them separately is advantageous. Consequently, we selectively apply the effects of $\mathcal{L}_{rec}$ to guide the learning of $g^i(.)$, while $\mathcal{L}_{P\text{-}U}$ guides $f^i(.)$, $f^0(.)$, and $g^0(.)$ for $i = 1, \ldots, M$. To fully leverage label information to learn Robult's task head $c(.)$, we introduce an additional supervised loss $\mathcal{L}_{sup}$ on labeled data, directing the learning process of the entire Robult network. Depending on the task (regression or classification), we utilize well-established $L_1$ or cross-entropy $\mathcal{L}_{ce}$ losses. A detailed procedure is available in Appendix - A.3.

## 3 EXPERIMENTAL RESULTS

### 3.1 DATASETS AND METRICS

**Dataset:** We conduct experiments on the following datasets. *CMU-MOSI* Zadeh et al. (2016) & *CMU-MOSEI* Zadeh et al. (2018b): These two datasets consist of three modalities (textual, sound,

and visual) all extracted from videos. They are associated with sentiment analysis and emotion recognition tasks. All videos are labeled on a scale ranging from -3 (negative sentiment) to 3 (positive sentiment). *MM-IMDb* Arevalo et al. (2017): This dataset is designed for a genre classification task with image and text modalities. The task involves multi-label classification since a movie might have several genres. *UPMC Food-101* Wang et al. (2015): This dataset is a classification dataset consisting of 101 food categories. There are two modalities, text and images, collected from Google Image Searches. *Hateful Memes* Kiela et al. (2020): This dataset aims at identifying hate speech in memes via text and image modalities. Created by Meta AI, this dataset includes challenging examples that are similar to hateful ones but are actually harmless.

**Metrics:** For sentiment analysis related to CMU-MOSI and CMU-MOSEI datasets, we adopt mean absolute error (MAE), correlation (Cor), binary accuracy, and F1 score, following Poklukar et al. (2022b); Tsai et al. (2018). Here, binary categories determine positive sentiment scores ($> 0$) or negative ones ($< 0$). For the evaluation of the three remaining datasets, we adhere to the metrics specified in Lee et al. (2023b). With the MM-IMDb dataset, the multi-label classification performance is assessed using F1-Macro. The classification accuracy is employed for the UPMC Food-101 dataset. Lastly, for Hateful Memes, the evaluation is based on the AUROC metric.

## 3.2 Baselines and Experimental Settings

**Baselines:** We incorporate several state-of-the-art approaches representing popular strategies into our comparative evaluation. Specifically, GMC Poklukar et al. (2022b) serves as a contrastive learning-based approach, ActionMAE Woo et al. (2023) represents a generation-based method, and we include a Transformer-based approach proposed in Lee et al. (2023b), referred to as Prompt-Trans for brevity. To ensure optimal reproducibility, we inherit the implementations of all baseline methods from their original code bases. Additionally, we implement unimodal frameworks (Unimodal) for each modality, trained in a supervised manner with available labels, to serve as our baseline comparison.

**Implementation details:** To ensure a fair comparison, we use similar encoder architectures for processing raw data modalities whenever possible. The unimodal baselines are designed with the same architectures as Robult, each with its own classifier. For Robult, positive samples for the soft P-U loss are determined after discretizing labels if needed. Specifically, in the cases of CMU-MOSI and CMU-MOSEI datasets, label information in the range of $[-3, 3]$ is quantified into 7 discrete categories $(-3, -2, \ldots, 3)$. Additionally, for the multi-label dataset MM-IMDb, two samples are considered positive if they share all the same labels. Regarding Prompt-Trans, we only report its results for three datasets involving two modalities (MM-IMDb dataset, UPMC Food-101 dataset, and Hateful Memes dataset), as the extension to multiple modalities cannot be directly inferred from the original work Lee et al. (2023b).

**Experimental details:** The primary focus of our performance reporting is on two extreme scenarios: semi-supervised settings with only $5\%$ labeled data and scenarios where only a single modality is presented during evaluation. All reported results are averaged over 3 different random seeds. In the semi-supervised setup, the newly created labeled portion is ensured to maintain the correct label ratio as the original training sets. Additional experiments extending these two settings to more modalities and a higher percent of labeled data are detailed in Appendix C.1 and C.2 respectively. Specific details on implementation settings relating to each dataset are provided in Appendix - B.2.

## 3.3 Main Quantitative Results

All results are shown in tables with the best outcomes in **red** and the second-best in blue.

**Sentiment Analysis:** The results for CMU-MOSI and CMU-MOSEI datasets are summarized in Table 1. For both datasets, Robult significantly outperforms all the compared methods, suggesting its effectiveness and consistency in semi-supervised and missing modality scenarios. Regarding CMU-MOSI, due to its smaller scale compared to CMU-MOSEI, the labeled portions are also smaller. This condition poses a challenge for existing baselines that heavily rely on label information. In contrast, Robult effectively addresses this challenge, demonstrating the ability to extract meaningful representations even with limited labeled data. On CMU-MOSEI, Robult consistently produces superior representations, achieving the best performances across all recorded metrics. Notably, Robult improves the correlation (Corr) between the predicted sentiment levels and ground truth by up to $19.8\%$, outperforming the second-best method, which is the unimodal for textual data.

Table 1: Results on CMU-MOSI, CMU-MOSEI.

| | CMU-MOSI | | | | CMU-MOSEI | | | |
|---|---|---|---|---|---|---|---|---|
| Metrics | Unimodal | GMC | ActionMAE | Robult | Unimodal | GMC | ActionMAE | Robult |
| *Text Modality:* | | | | | | | | |
| MAE ($\downarrow$) | 1.41 | 1.407 | 1.476 | **1.397** | 0.81 | 0.815 | 1.115 | **0.784** |
| Corr ($\uparrow$) | 0.137 | 0.14 | 0.066 | **0.144** | 0.383 | 0.346 | 0.136 | **0.459** |
| F1 ($\uparrow$) | 0.551 | 0.559 | 0.535 | **0.578** | 0.717 | 0.716 | 0.614 | **0.739** |
| Acc ($\uparrow$) | 0.553 | 0.562 | 0.47 | **0.569** | 0.712 | 0.708 | 0.603 | **0.732** |
| *Audio Modality:* | | | | | | | | |
| MAE ($\downarrow$) | 1.576 | 1.518 | 1.546 | **1.415** | 0.842 | 0.836 | 1.215 | **0.825** |
| Corr ($\uparrow$) | 0.041 | -0.065 | 0.046 | **0.085** | 0.111 | 0.193 | 0.101 | **0.221** |
| F1 ($\uparrow$) | 0.512 | 0.457 | 0.508 | **0.539** | 0.618 | 0.642 | 0.634 | **0.679** |
| Acc ($\uparrow$) | 0.496 | 0.46 | 0.467 | **0.535** | 0.599 | 0.63 | 0.543 | **0.65** |
| *Vision Modality:* | | | | | | | | |
| MAE ($\downarrow$) | 1.451 | 1.497 | 1.511 | **1.425** | 0.891 | 0.839 | 1.127 | **0.826** |
| Corr ($\uparrow$) | 0.044 | -0.07 | -0.03 | **0.086** | 0.163 | 0.2 | 0.104 | **0.201** |
| F1 ($\uparrow$) | 0.585 | 0.446 | 0.511 | **0.593** | 0.637 | 0.621 | 0.594 | **0.647** |
| Acc ($\uparrow$) | 0.425 | 0.449 | 0.514 | **0.522** | 0.624 | 0.62 | 0.561 | **0.632** |
| *Full Modality:* | | | | | | | | |
| MAE ($\downarrow$) | 1.394 | 1.47 | 1.496 | **1.392** | 0.783 | 0.819 | 1.103 | **0.779** |
| Corr ($\uparrow$) | 0.186 | 0.101 | -0.092 | **0.247** | 0.364 | 0.328 | 0.337 | **0.504** |
| F1 ($\uparrow$) | 0.597 | 0.497 | 0.553 | **0.657** | 0.73 | 0.693 | 0.694 | **0.744** |
| Acc ($\uparrow$) | 0.594 | 0.498 | 0.477 | **0.63** | 0.729 | 0.688 | 0.643 | **0.741** |

Table 2: Results on MM-IMDb, UPMC Food-101, Hateful Memes.

| | Unimodal | Prompt-Trans | GMC | ActionMAE | **Robult** |
|---|---|---|---|---|---|
| *MM-IMDb - F1 Macro ($\uparrow$):* | | | | | |
| Text | 0.24 | 0.198 | 0.296 | 0.055 | **0.321** |
| Image | 0.207 | 0.148 | 0.291 | 0.039 | **0.298** |
| Full | 0.196 | 0.268 | 0.307 | 0.171 | **0.332** |
| *UPMC Food-101 - Accuracy ($\uparrow$):* | | | | | |
| Text | 0.321 | 0.151 | 0.395 | 0.196 | **0.435** |
| Image | 0.296 | 0.111 | 0.382 | 0.132 | **0.415** |
| Full | 0.138 | 0.432 | 0.41 | 0.358 | **0.446** |
| *Hateful Memes - AUROC ($\uparrow$):* | | | | | |
| Text | 0.584 | 0.511 | 0.617 | 0.528 | **0.623** |
| Image | 0.524 | 0.475 | 0.528 | 0.508 | **0.596** |
| Full | 0.618 | **0.635** | 0.616 | 0.542 | 0.632 |

Figure 4: CD diagram showing the mean rank of each method on three datasets.

**Classification tasks:** In Table 2, empirical results for three classification tasks show that Robult consistently outperforms existing approaches and baselines in most cases, except for one scenario on the Hateful Memes dataset with the full modality available, where Robult achieves comparable performance with Prompt-Trans Lee et al. (2023b). Notably, the Hateful Memes dataset includes samples with "benign confounders", negatively impacting performance when models rely solely on single modalities Kiela et al. (2020). Leveraging the soft Positive-Unlabelled loss, Robult effectively addresses and mitigates performance gaps with either single modality inputs or the full ones. In addition, we calculate F1 macro scores for all methods on these three datasets in the unimodal and multimodal cases. We further visualize a Critical Difference Diagram Demšar (2006) in Figure 4. This diagram visually represents the performance among different machine learning algorithms across various datasets by displaying the mean performance ranks, with lower being better, and connecting statistically indistinguishable groups (within $95\%$ confidence level) with a thin horizontal bar, as per the Friedman hypothesis test. From the diagram, Robult exhibits a clear improvement gap compared to other state-of-the-art methods in average ranks, while ActionMAE and Prompt-Trans show no statistically significant difference in their performance.

### 3.4 ADDITIONAL QUALITATIVE ANALYSIS

In this section, we provide two key analyses: an evaluation of the quality of the learned representations and the main ablation study of Robult. For a more comprehensive analysis, please refer to the additional experiments in Appendix C, which offer further insights into how architectural choices, Soft-PU loss, and weighting schemes influence Robult's performance.

**Alignment and Uniformity:** We assess the learned representations $Z^i$ and $S$ after the trainning process with soft P-U loss, via two qualities - Alignment and Uniformity Wang & Isola (2020). Figure 5 provides a comprehensive analysis of the learned representations generated by Robult using both unimodal and multimodal inputs on the Hateful Memes testing set. On the left, the Frobenius-norm distance histograms of positive pairs within the test dataset indicate that the representations generated with all the modalities have the smallest mean distances, and as the distances increase, their corresponding density decreases. While not as compact as the representations with full modalities input, positive pairs' representations generated with unimodal input still exhibit low mean distances and good histogram shapes. Furthermore, to analyze the uniformity characteristics of the learned representations, we follow the process outlined in Wang & Isola (2020) and show the result on the right of Figure 5. The learned representations are projected into $\mathbb{R}^2$ using t-SNE Van der Maaten & Hinton (2008), and the output feature distributions are visualized using Gaussian kernel density estimation (KDE) along with von Mises-Fisher (vMF) KDE for angles (`arctan2(y;x)`). As suggested by these figures, Robult's representations demonstrate uniform characteristics on the entire test set as well as good clustering between classes. Specifically, representations of different classes

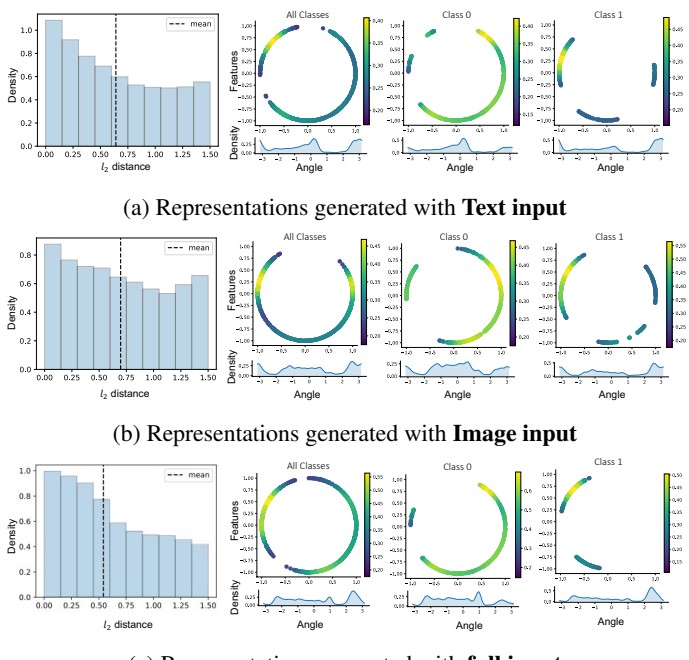

(a) Representations generated with **Text input**

(b) Representations generated with **Image input**

(c) Representations generated with **full input**

Figure 5: Alignment and Uniformity analysis on representations of Hateful Memes test dataset, generated by Robult.

reside on different segments of the unit circle and form separated clusters in Figure 9. The level of separation for different classes with different input modalities correlates well with the actual quantitative results, as shown in Table 2. Additional clustering comparison between different method can be found in Appendix C.8

Table 3: Ablation analysis on CMU-MOSI and Hateful Memes datasets for Robult.

| Metrics | GMC | **Robult** | Robult (1) | Robult (2) | Robult (3) | Robult (4) |
|---------|-----|--------|-----------|-----------|-----------|-----------|
| | *CMU-MOSI - Text Modality:* | | | | | |
| MAE | 1.407 | **1.397** | 1.589 | 1.511 | 1.443 | 1.429 |
| Corr | 0.14 | **0.144** | 0.047 | 0.101 | 0.051 | 0.123 |
| F1 | 0.559 | 0.578 | 0.542 | 0.52 | **0.593** | 0.571 |
| Acc | 0.562 | 0.569 | 0.544 | 0.523 | 0.422 | **0.573** |
| | *CMU-MOSI - Audio Modality:* | | | | | |
| MAE | 1.518 | **1.415** | 1.586 | 1.561 | 1.494 | 1.495 |
| Corr | -0.065 | **0.085** | 0.023 | 0.005 | 0.046 | **0.085** |
| F1 | 0.457 | **0.539** | 0.526 | 0.499 | 0.51 | 0.517 |
| Acc | 0.46 | **0.535** | 0.518 | 0.502 | 0.442 | 0.509 |
| | *CMU-MOSI - Vision Modality:* | | | | | |
| MAE | 1.497 | **1.425** | 1.663 | 1.711 | 1.445 | 1.504 |
| Corr | -0.07 | **0.086** | 0.025 | -0.023 | 0.041 | -0.066 |
| F1 | 0.446 | **0.593** | 0.519 | 0.485 | 0.571 | 0.459 |
| Acc | 0.449 | **0.522** | 0.519 | 0.465 | 0.47 | 0.448 |
| | *CMU-MOSI - Full Modality:* | | | | | |
| MAE | 1.47 | **1.392** | 1.588 | 1.434 | 1.411 | 1.459 |
| Corr | 0.101 | **0.247** | 0.071 | 0.239 | 0.166 | 0.229 |
| F1 | 0.497 | **0.657** | 0.524 | 0.567 | 0.549 | 0.6 |
| Acc | 0.498 | **0.63** | 0.523 | 0.566 | 0.552 | 0.601 |

| Metrics | GMC | **Robult** | Robult (1) | Robult (2) | Robult (3) | Robult (4) |
|---------|-----|--------|-----------|-----------|-----------|-----------|
| | *Hateful Memes - Text Modality:* | | | | | |
| AUROC | 0.617 | **0.623** | 0.528 | 0.59 | 0.605 | 0.576 |
| Acc | 0.581 | **0.59** | 0.535 | 0.556 | 0.571 | 0.562 |
| | *Hateful Memes - Image Modality:* | | | | | |
| AUROC | 0.528 | **0.596** | 0.518 | 0.582 | 0.588 | 0.566 |
| Acc | 0.551 | **0.562** | 0.524 | 0.551 | 0.526 | 0.539 |
| | *Hateful Memes - Full Modality:* | | | | | |
| AUROC | 0.616 | 0.632 | 0.538 | 0.618 | **0.634** | 0.582 |
| Acc | 0.532 | **0.595** | 0.542 | 0.55 | 0.554 | 0.552 |

**Ablation Studies:** We evaluate the impact of each loss component on Robult's performance using CMU-MOSI and Hateful Memes datasets, which mirror the semi-supervised and missing modalities conditions of our main experiments. This analysis involves testing variations of Robult with different ablations. (1) *Removal of* $\mathcal{L}_{sup}$ - this setting utilizes available label information only in $\mathcal{L}_{(u)lb}$, so Robult can only produce latent representations. An additional Logistic Regressor is trained with these representations as its input, and this pipeline's final scores are reported. (2) *Removal of* $\mathcal{L}_{rec}$ - this setting discards $\mathcal{L}_{rec}$, corresponding to our Objective 2.2. (3) *Removal of* $\mathcal{L}_{lb}$ - this setting makes the learning of Objective 2.1 rely only on $\mathcal{L}_{ulb}$. (4) *Removal of* $\mathcal{L}_{ulb}$ - this setting associates Objective 2.1 exclusively with $\mathcal{L}_{lb}$. Table 3 summarizes the results of this ablation experiment.

Overall, any ablation negatively impacts the performance of Robult. In particular, the absence of $\mathcal{L}_{sup}$ significantly worsens the performance, as there is no loss guiding the learning of Robult's classifier, which is crucial for generating soft label information consumed by the soft Positive-Unlabeled loss $\mathcal{L}_{ulb}$. Consequently, this ablation adversely affects two loss components, explaining the poorest result among all variations. The removal of $\mathcal{L}_{rec}$ particularly harms the performance with unimodal inputs, aligning with the motivation for Objective 2.2, as the unique information $U^i$ is no longer preserved. In two remaining cases, both ablations diminish Robult's overall performance, indicating their equal contribution to achieving Objective 2.1.

## 4 RELATED WORKS

**Semi-supervised Multimodal Learning:** Several works acknowledge the challenge of fully labeled datasets in the multimodal literature and provide targeted solutions for specific applications Lian et al. (2022); Zheng et al. (2022); Zhang et al. (2023); Liang et al. (2023). For instance, in Aberdam et al. (2022), the authors tackle the semi-supervised scenario in scene text recognition by enforcing consistency between weakly augmented pseudo-labels and strongly augmented views. SMIN Lian et al. (2022) addresses conversational emotion recognition tasks through intra-modal and cross-modal interactive modules inspired by auto-encoders. In Zheng et al. (2022), labeled hash codes are learned using label signals, preserving the data structure of unlabeled ones, followed by importance differentiation regression for final multimodal hashing. Authors in Zhang et al. (2023) propose an area-similarity contrastive loss for medical image segmentation, leveraging cross-modal information to enhance representations of unlabeled data. Liang et al. Liang et al. (2023) derive two lower bounds of multimodal interaction from an information-theoretic perspective, applicable for pre-analysis of multimodal interaction effects. However, these efforts primarily focus on semi-supervised scenarios in specific tasks and certain modalities (e.g., text-images), limiting their applicability to general cases. A common technique in general semi-supervised learning is loss reweighting based on pseudolabel uncertainty, similar to our P-U loss. These methods aim to mitigate confirmation bias and are widely used in unsupervised domain adaptation Li et al. (2021); Litrico et al. (2023), particularly in image processing applications Jin et al. (2022); Lee et al. (2023a). However, such methods have not been applied to multimodal scenarios, where the complexity of loss reweighting increases. This distinguishes our P-U loss method.

**Missing modalities:** Many multimodal fusion methods rely on a complete set of modalities, but deployment settings often lack such ideal conditions, leading to adverse effects when using these strategies Wang et al. (2020); Ma et al. (2022). To address this challenge, some approaches aim to create models resilient to missing modalities Ma et al. (2021; 2022); Poklukar et al. (2022b); Woo et al. (2023); Lee et al. (2023b). For instance, Wang et al. Wang et al. (2020) optimize training by considering incomplete data samples to generate unimodal teachers guiding a multimodal student. Smil Ma et al. (2021) approximates latent features of modality-incomplete data using Bayesian meta-learning. GMC Poklukar et al. (2022b) preserves geometric alignment in multimodal representations, enabling unimodal representations to substitute for absent representations of other modalities. ActionMAE, inspired by the masked autoencoder idea Feichtenhofer et al. (2022); Bachmann et al. (2022), learns to predict the latent representation of a missing modality by randomly dropping its feature token and learning to reconstruct it. Despite success in certain scenarios, these frameworks often rely on labeled signals, implicitly or explicitly, in the training dataset, limiting their general applicability.

## 5 CONTRIBUTIONS & LIMITATIONS

**Contributions:** Our Robult pipeline effectively uses limited label data through a soft Positive-Unlabelled (P-U) loss and latent reconstruction loss, enhancing modality interactions and preserving unimodal data integrity. It supports various modality types and quantities, scales linearly with modalities, and functions independently of specific encoders/decoders. This flexibility facilitates integration with existing deep learning frameworks, advancing multimodal learning in practical settings. **Limitations.** Robult's design presumes that the proximity of positive couplets follows a Gaussian distribution, a method proven empirically but not theoretically. Future work should seek theoretical validation for this assumption. Moreover, while focused on semi-supervised environments with complete modalities during training, the potential of labeled data in scenarios with missing modalities in training remains untapped. Exploring these cases could further improve Robult's effectiveness in complex real-world applications.

## REPRODUCABILITY STATEMENT

We have made extensive efforts to ensure the reproducibility of our work, focusing on several key areas:

- Code Availability: The complete codebase for this work, including models, training scripts, and evaluation procedures, is uploaded as the Supplementary material. Upon acceptance, this code will be open-sourced and made publicly available on GitHub.

- Dataset Preparation: Detailed instructions for dataset setup, including any preprocessing steps and data splits used in our experiments, are provided in Section 3, Appendix B.2, enabling other researchers to replicate our exact experimental conditions.

- Hardware and Hyperparameters: A comprehensive description of the hyperparameters used in our experiments, including optimization settings, the GPUs used, and other configuration details, is provided in Appendix B.1, B.2.

- Architecture Transparency: Detailed descriptions of our model architectures are provided in Appendix B.2, ensuring others can understand and reconstruct the models accurately.

- Evaluation Metrics: The exact definitions of all evaluation metrics used are provided in Section 3 of the main paper.

By offering this comprehensive set of resources, we aim to facilitate the reproduction of our results by the research community. We believe that this level of transparency is crucial for advancing the field and supporting thorough validation and extension of our work.

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

## A ROBULT SUPPLEMENTARY DETAILS

### A.1 MINIMIZING CONDITIONAL ENTROPY WITH LATENT RECONSTRUCTION ERROR

As explained in the primary text, our approach to achieve Objective 2.2 involves a reconstruction procedure with two components: the reconstruction module $r^i(U^i, Z^i) = \tilde{H}^i$ and the latent reconstruction loss $\mathcal{L}_{rec}$. This procedure is illustrated in Figure 6. It is important to note that these

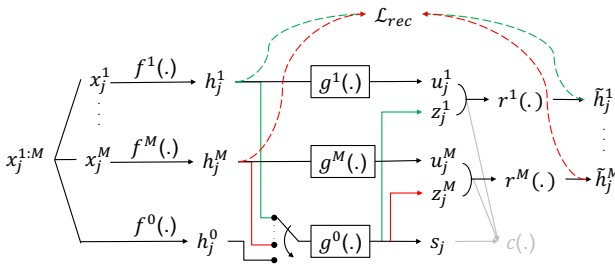

Figure 6: Reconstruction procedure of Robult. This procedure only applied in training stage.

reconstruction modules $r^i(.)$ are used exclusively during the learning process to optimize individual branches $g^i(.)$, incurring no additional overhead during the evaluation or deployment stages. As this reconstruction is carried out in the latent space, the module $r^i(.)$ can be uniformly designed, irrespective of the characteristics of input modalities. In our Robult design, $r^i(.)$ is simply a two-layered MLP with ReLU activation in the middle, applied across all five datasets.

### A.2 MAXIMIZING MUTUAL INFORMATION WITH SOFT POSITIVE-UNLABELED CONTRASTIVE LEARNING

In this section, we would derive the lower bound of mutual information between synergy latent $S$ and unimodal representation $Z^i$ as a Positive-Unlabelled learning objective, which relax the assumption about full presence of labels in training dataset. This derivation explains Result 2 in the main manuscript.

The ultimate goal is to maximize the following MI quantity:

$$\mathcal{MI}(S, Z^i) = d_{KL}\left(p_{S,Z^i}||p_S \otimes p_{Z^i}\right)$$

This essentially means that the KL divergence between the joint distribution $p_{S,Z^i}$ and the product of marginal distribution $p_S \otimes p_{Z^i}$ should be maximized. As defined in the main manuscript, $F$ is the flag indicator denotes whether a couplet $(s, z^i)$ is sampled from the joint distribution $p_{S,Z^i}$ ($F = 1$) or from product of marginal distribution $p_S \otimes p_{Z^i}$ ($F = 0$):

$$p(S, Z^i|F = 1) = p_{S,Z^i}; \;\; p(S, Z^i|F = 0) = p_S \otimes p_{Z^i}; \tag{11}$$

Applying Bayes' rule, the posterior for $F = 1$ is given by:

$$
\begin{aligned}
p(F = 1|S, Z^i) &= \frac{p(S, Z^i|F = 1)p(F = 1)}{p(S, Z^i)} \\
&= \frac{p(S, Z^i|F = 1)p(F = 1)}{p(S, Z^i|F = 1)p(F = 1) + p(S, Z^i|F = 0)p(F = 0)} \\
&= \frac{p_{S,Z^i} \cdot p(F = 1)}{p_{S,Z^i} \cdot p(F = 1) + p_S \otimes p_{Z^i} \cdot p(F = 0)}.
\end{aligned}
\tag{12}
$$

Putting logarithm operation on both side of Equation 12:

$$
\begin{aligned}
\log p(F = 1|S, Z^i) &= -\log\left(1 + k\frac{p_S \otimes p_{Z^i}}{p_{S,Z^i}}\right) \\
&\leq -\log k + \log\frac{p_{S,Z^i}}{p_S \otimes p_{Z^i}},
\end{aligned}
\tag{13}
$$

in which

$$k = \frac{p(F = 0)}{p(F = 1)}. \tag{14}$$

Taking expectation w.r.t $p_{S,Z^i}$ (or $p(S, Z^i|F = 1)$), we can bound the mutual information as

$$\mathcal{MI}(S, Z^i) \geq \mathbb{E}_{p(S,Z^i|F=1)}\log p(F = 1|S, Z^i) + \log k \tag{15}$$

Here, the true distribution $p(F = 1|S, Z^i)$ is unknown, so we approximate it with a well-established non-parametric model $v : S \times Z^i \to [0, 1]$ Chen et al. (2021b; 2023):

$$\mathcal{MI}(S, Z^i) \geq \mathbb{E}_{p(S,Z^i|F=1)}\log v(S, Z^i) + \log k$$

where

$$v(s_j, z_j^i) = \frac{\phi(s_j, z_j^i)}{\sum_k^B \phi(s_j, z_k^i)}; \tag{16}$$
$$\phi(s_j, z_j^i) = exp(< s_j; z_j^i > /\tau).$$

In addition, let $c$ be the number of underlying classes and assume the labels are uniformly distributed, we have the probability that a couplet is sharing a label is $p(f_k = 1) = \frac{1}{c^2}$. Within mini-batch of size $B$, consider the scenario in which the number of positive couplets $B_p$ is greater than the number of negative ones $B_n$ (hence $B_p > \frac{\binom{B}{2}}{2} = \frac{\tilde{B}}{2}$):

$$p(B_p) = \binom{\tilde{B}}{B_p} \cdot \frac{1}{c^{2B_p}}$$
$$\leq \binom{\tilde{B}}{\frac{\tilde{B}}{2}} \cdot \frac{1}{c^{\tilde{B}}}$$

It should be noted that this possibility $p(B_p)$ is upper-bounded by a small quantity given $B > 1$ and $c \geq 2$ (smaller than $0.1$ in case $B = 8$ and $c = 2$), and get smaller when $B$ and $c$ increase. Intuitively, the possibility that the positive couplets outnumber the negative ones is negligible, hence, it normally hold true that:

$$\log k = \log \frac{p(F = 0)}{p(F = 1)} \geq 0.$$

With this realization, Result 2 can be derived from Eq. 16 as follow:

$$\begin{aligned} I(S, Z^i) &\geq \mathbb{E}_{p(S,Z^i|F=1)}\log v(S, Z^i) + \log k \\ &\geq \mathbb{E}_{p(S,Z^i|F=1)}\log v(S, Z^i) \\ &= \mathbb{E}_{p_{S,Z^i}}\log v(S, Z^i). \end{aligned} \tag{17}$$

## A.3 TRAINING STRATEGY

We employ an end-to-end training pipeline that can process two objectives 2.2 and 2.1 independently, as demonstrated by Algorithm 1. In general, we selectively perform gradient calculations on different modules of Robult based on the specific losses. This selection process offers dual benefits: (1) minimal processing overhead with a single forward pass, and (2) effective restriction of the losses' impact only on their target modules.

## A.4 ROBULT'S COMPLEXITY ANALYSIS

Robult framework is built on two main types of modules: individual branch modules - $g^i(.)$ ($i = \overline{0, M}$), and reconstruction modules - $r^i(.)$ ($i = 1, \ldots, M$). For the projectors and the fusion module, denoted as $f^i(.)$ ($i = \overline{0, M}$), we adopt designs from previous studies. In this section, we analyze the complexity of our two proposed modules, which operate in latent spaces and have straightforward designs.

### A.4.1 INDIVIDUAL BRANCHES

Since all $g^i(.)$'s are working with the same input latent space (all raw modalities are projected to the same space), we unify the design of $g^i(.)$'s to be identical across different modalities. Specifically, $g^i(.)$ are constituted by multiple Fully Connected layers, with middle $ReLU$ activations; the last layer of $g^i(.)$ involve no activation, but a L2 normalization operation. Below are the table of hyperparameters involved in the analysis.

**Algorithm 1** Robult training strategy

**Input:**
▷ Training dataset $\mathcal{D}_{train}$
▷ Robult framework $\mathcal{RB}$
▷ Optimizer $\mathcal{O}$
**function** *ParametersToggle*($f$: *flag_variable*)
   **if** $f = 0$ **then**
      Toggle all $\mathcal{RB}$ parameters to require gradient calculation
   **else if** $f = 1$ **then**
      Toggle all $\mathcal{RB}$ parameters to **NOT** require gradient calculation
      Toggle all $g^i(.)$ parameters $(i = 1, \ldots, M)$ to require gradient calculation
   **else if** $f = 2$ **then**
      Toggle all $\mathcal{RB}$ parameters to **NOT** require gradient calculation
      Toggle all $f^i(.), g^i(.)$ parameters $(i = \overline{0, M})$ to require gradient calculation
   **end if**
**end function**
**for** $B_i; Y_i$ **in** $\mathcal{D}_{train}$ **do**
   ▷ *single forward pass*
   $\tilde{Y}_i, H_i, Z_i, U_i, S = \mathcal{RB}(B_i)$
   ▷ *loss calculations*
   $l_{cls} = \mathcal{L}_{cls}(\tilde{Y}_i, Y_i)$
   $l_{rec} = \mathcal{L}_{rec}(H_i, Z_i, U_i)$
   $l_{(u)lb} = \mathcal{L}_{(u)lb}(Z_i, S)$
   ▷ *gradient calculations*
   Call *ParametersToggle*($f = 1$); backward with $l_{cls}$
   Call *ParametersToggle*($f = 2$); backward with $l_{lb}$ and $l_{ulb}$
   Call *ParametersToggle*($f = 0$); backward with $l_{sup}$
   ▷ *single backward pass*
   Optimizer $\mathcal{O}$ update $\mathcal{RB}$ parameters with above gradient infomation
**end for**

Table 4: $g^i(.)$ related hyper-parameters

| Notation | Description |
|---|---|
| $M$ | number of modalities |
| $L$ | number of FC layers |
| $d_i$ | hidden dimension of $i^th$ layer's output |
| $d_0$ | input dimension |

**Time Complexity.** Assume a single operation can be performed in unit time ($\mathcal{O}(1)$). We have the calculation for number of operations in a forward pass as follows.

Within the $i^th$ FC layer:
$$d_{i-1} * d_i + di,$$
Over $L$ layers:
$$\sum_{i=1}^{L} d_{i-1} * d_i + di.$$

In our implementations, we choose the same dimensions for all hidden outputs (same $d = d_i \forall i = \overline{1, L}$), and there are $M + 1$ modules $g^i(.)$. With this, the total number of operation is:

$$(M + 1)\sum_{i=1}^{L} d_{i-1} * d_i + di = (M + 1) * L * d * (d + 1) = \mathcal{O}(M * L * d^2)$$

By utilizing matrix product and GPU acceleration, $d^2$ operations can in fact be performed in $\mathcal{O}(1)$ time, make the whole time complexity for individual branches be $\mathcal{O}(M * L)$, which is linearly scaled with $M$.

**Space Complexity.** Regarding the space complexity, within $i^{th}$ layer, beside the need for storing parameter matrix of size $(d_{i-1} + 1) \times d_i$, output after performing $ReLU$ activation are also stored to later perform back-propagation. Hence, the total number of stored parameters is:

$$(d_{i-1} + 1) * d_i + d_i = (d_{i-1} + 2) * d_i.$$

Following similar derivation with $L$ layers and $M + 1$ branches, replacing $d = d_i \forall i = \overline{1, L}$, we have the total space complexity is:

$$(M + 1) * L * (d + 2) * d = \mathcal{O}(M * L * d^2).$$

### A.4.2 RECONSTRUCTION MODULES

For these reconstruction modules $r^i(.)$'s, we also adopt a similar design patterns as that of individual branches $g^i(.)$. The only differences are the dimension of input for first FC layer $(2d)$, which corresponding to the concatenation of $g^0(.)$'s and $g^i(.)$'s outputs.

**Time complexity.** With that intuition, as we have $M$ branches and $L$ layers, the total number of calculations is:

$$M * [(2d + 1) * d + (L - 1) * (d + 1) * d] = M * \left[d^2 + L * (d + 1) * d\right] = \mathcal{O}(M * L * d^2)$$

Reducing $d^2$ operations to $\mathcal{O}(1)$ time complexity, the same result as observed with $g^i(.)$'s are observed - $\mathcal{O}(M * L)$.

**Space complexity.** The total number of stored parameters is:

$$M * [(2d + 2) * d + (L - 1) * (d + 2) * d] = M * \left[(d^2 + L * (d + 2) * d\right] = \mathcal{O}(M * L * d^2).$$

In conclusion, all the proposed modules of Robult are linearly scaled (both in time and space), with the number of modalities $M$.

### A.4.3 COMPUTATIONAL TIME QUANTITATIVE RESULT

Table 5: Computational times of different methods on different datasets.

| Robult | GMC | ActionMAE |
|---|---|---|
| *CMU-MOSI:* | | |
| 1.08 GFLOPS | 1.05 GFLOPS | 1.13 GFLOPS |
| 507.44 MMACs | 492 MMACs | 526.36 MMACs |
| 1.46 M | 1.16 M | 11.55 M |
| *CMU-MOSEI:* | | |
| 1.09 GFLOPS | 1.06 GFLOPS | 1.15 GFLOPS |
| 508.7 MMACs | 493.26 MMACs | 526.36 MMACs |
| 1.41 M | 1.17 M | 11.56 M |
| *Hateful Memes:* | | |
| 32.81 MFLOPs | 26.56 MFLOPS | 61.37 MFLOPS |
| 16.39 MMACs | 13.27 MMACs | 30.48 MMACs |
| 888.32 K | 674.05 K | 1.04 M |

To further substantiate our results, we measured FLOPs and MACs for several datasets we utilized, comparing them with our current baselines (Table 5). For a fair comparison, we kept all unimodal projectors the same. The results suggest that our method introduces slight overheads compared to GMC, but remains faster than ActionMAE, while significantly outperforming both methods in downstream performance.

## B IMPLEMENTATION DETAILS

### B.1 ENVIRONMENT SETTINGS

All implementations and experiments are conducted on a single machine equipped with the following hardware configuration: a 6-core Intel Xeon CPU paired with 2 NVIDIA A100 GPUs for accelerated training.

Our codebase predominantly utilizes the *PyTorch 2.0* framework, including the *Pytorch-AutoGrad*, for deep learning model design and calculations. Additionally, we leverage utilities from *Scikit-learn, Pandas,* and *Matplotlib* to support various functionalities in our experiments. The original codebase for Robult will be made publicly available upon publication.

In both Robult and the Unimodal baselines, we modify the architecture of the projectors $f^i(.)$ ($i = \overline{0, M}$) while keeping $g^i(.)$ ($i = \overline{0, M}$) as simple as possible. For all testing datasets, the unimodal branches $g^i(.)$ consist of a simple Fully Connected layer followed by $L_2$ normalization. In contrast, $g^0(.)$ has a higher representation capacity with two Fully Connected layers and ReLU activation. In the case of GMC Poklukar et al. (2022b) and ActionMAE Woo et al. (2023), the same architecture of the projectors is adopted as Robult to ensure a fair comparison, with the remaining designs being directly inherited from the original codebases. For Prompt-Trans Lee et al. (2023b), we keep all the architecture designs intact and only change the datasets' settings to semi-supervised and missing modalities scenarios. Additional information about the baselines should be best referenced from their original works.

**CMU-MOSI and CMU-MOSEI datasets.** We follow the settings in Poklukar et al. (2022b), which involve temporally-aligned versions of these datasets generated with Zadeh et al. (2018a), with additional adaptions for semi-supervised and missing modalities scenarios. Specifically, multimodal Transformer Tsai et al. (2019) is adopt as the joint-modality encoder $f^0(.)$ for our model and all state-of-the-art baselines; single-layer GRUs are adopted as unimodal projectors $f^i(.)$. The latent space dimension is set as $60$, and all methods are trained in $40$ epochs with Adam optimizer Kingma & Ba (2014) at the learning rate of $10^{-3}$.

**MM-IMDb, UPMC Food-101 and Hateful Memes datasets.** For the classification tasks associated with the MM-IMDb, UPMC Food-101, and Hateful Memes datasets, we initially generate text and visual embeddings offline using a pretrained ViLT framework Kim et al. (2021). Subsequently, all models are trained using these embeddings instead of raw data. This specific procedure is intentionally conducted to ensure fair evaluation, as Prompt-Trans Lee et al. (2023b) functioning also involves the same frozen ViLT framework in their training process. We also follow Prompt-Trans Lee et al. (2023b) for the preprocessing procedures of raw texts and images. For all methods, the offline embedding space's dimension is fixed at $784$ and further condensed into a $128$-dimensional hidden latent space. Additionally, with this setting, we choose the projectors $f^i(.)$ ($i = \overline{0, M}$) as simple Fully Connected layers.

## C  Additional empirical results and analysis

In this sections, we present additional empirical results and analysis to study the behavior of Robult and baselines in different extended settings.

### C.1  Extended modalities missing scenarios

In Table 6, we present the comprehensive performance of different frameworks when provided access to all combinations of input modalities on CMU-MOSI and CMU-MOSEI datasets. This table extends the information presented in Table 1 in the main text. In this experiment, to report the performances of Unimodal baselines and Robult when provided with two modalities, we simply take the mean of the outputs generated by providing these frameworks with single modalities. It is important to note that we do not draw conclusions on the best strategy for merging unimodal results. Despite this, using this simple strategy, Robult consistently produces the best results in most scenarios, highlighting its performance consistency across different missing modality scenarios.

### C.2  Extended semi-supervised scenarios

This experiment is designed to observe the behavior of models when exposed to varying amounts of labeled information. In addition to the $5\%$ labeled ratio setting covered in the main text, we additionally evaluate all methods with $50\%$ labeled ratio and ideal supervised settings:

- **Semi-supervised learning with 50% labelled data.** Table 7 summarizes the results of the $50\%$ labeled setting with CMU-MOSI and Hateful Memes datasets. As indicated, all methods effectively leverage the increased label signal, resulting in improved performance. However, it's noteworthy that Robult demonstrates its superiority by outperforming other methods on both datasets across various metrics.

- **Supervised learning.** With this scenario, we evaluate all methods in an ideal case where fully labelled training dataset is available. Similar to the previous setting, all method further enhance their performance given more labeled data. Robult suggest the consistency by outperforming other baselines in most recorded metrics.

Table 6: Full performance of different frameworks on CMU-MOSI and CMU-MOSEI Dataset.

| | CMU-MOSI | | | | CMU-MOSEI | | | |
|---|---|---|---|---|---|---|---|---|
| Metrics | Unimodal | GMC | ActionMAE | Robult | Unimodal | GMC | ActionMAE | Robult |
| *Text Modality:* | | | | | | | | |
| MAE | 1.41 | 1.407 | 1.476 | **1.397** | 0.81 | 0.815 | 1.115 | **0.784** |
| Corr | 0.137 | 0.14 | 0.066 | **0.144** | 0.383 | 0.346 | 0.136 | **0.459** |
| F1 | 0.551 | 0.559 | 0.535 | **0.578** | 0.717 | 0.716 | 0.614 | **0.739** |
| Acc | 0.553 | 0.562 | 0.47 | **0.569** | 0.712 | 0.708 | 0.603 | **0.732** |
| *Audio Modality:* | | | | | | | | |
| MAE | 1.576 | 1.518 | 1.546 | **1.415** | 0.842 | 0.836 | 1.215 | **0.825** |
| Corr | 0.041 | -0.065 | 0.046 | **0.085** | 0.111 | 0.193 | 0.101 | **0.221** |
| F1 | 0.512 | 0.457 | 0.508 | **0.539** | 0.618 | 0.642 | 0.634 | **0.679** |
| Acc | 0.496 | 0.46 | 0.467 | **0.535** | 0.599 | 0.63 | 0.543 | **0.65** |
| *Vision Modality:* | | | | | | | | |
| MAE | 1.451 | 1.497 | 1.511 | **1.425** | 0.891 | 0.839 | 1.127 | **0.826** |
| Corr | 0.044 | -0.07 | -0.03 | **0.086** | 0.163 | 0.2 | 0.104 | **0.201** |
| F1 | 0.585 | 0.446 | 0.511 | **0.593** | 0.637 | 0.621 | 0.594 | **0.647** |
| Acc | 0.425 | 0.449 | 0.514 | **0.522** | 0.624 | 0.62 | 0.561 | **0.632** |
| *Text+Audio Modalities:* | | | | | | | | |
| MAE | 1.485 | 1.442 | 1.521 | **1.401** | 0.765 | 0.813 | 1.007 | **0.762** |
| Corr | 0.131 | 0.05 | 0.089 | **0.141** | 0.418 | 0.352 | 0.202 | **0.439** |
| F1 | 0.528 | 0.491 | 0.507 | **0.563** | 0.729 | 0.728 | 0.624 | **0.733** |
| Acc | 0.512 | 0.493 | 0.508 | **0.546** | 0.713 | 0.671 | 0.62 | **0.717** |
| *Text+Vision Modalities:* | | | | | | | | |
| MAE | 1.486 | 1.465 | 1.489 | **1.415** | 0.861 | 0.822 | 1.003 | **0.788** |
| Corr | 0.144 | 0.044 | 0.086 | **0.146** | 0.325 | 0.325 | 0.198 | **0.399** |
| F1 | 0.514 | 0.487 | 0.501 | **0.58** | 0.718 | **0.722** | 0.623 | 0.718 |
| Acc | 0.492 | 0.491 | 0.506 | **0.534** | 0.688 | 0.687 | 0.619 | **0.704** |
| *Audio+Vision Modalities:* | | | | | | | | |
| MAE | 1.432 | 1.499 | 1.534 | **1.426** | 0.824 | 0.824 | 1.173 | **0.812** |
| Corr | 0.014 | -0.075 | -0.035 | **0.091** | 0.214 | 0.233 | 0.147 | **0.244** |
| F1 | 0.492 | 0.445 | 0.55 | **0.581** | **0.748** | 0.663 | 0.637 | 0.663 |
| Acc | 0.486 | 0.448 | 0.453 | **0.527** | 0.638 | 0.633 | 0.623 | **0.64** |
| Full Modalities: | | | | | | | | |
| MAE | 1.394 | 1.47 | 1.496 | **1.392** | 0.783 | 0.819 | 1.103 | **0.779** |
| Corr | 0.186 | 0.101 | -0.092 | **0.247** | 0.364 | 0.328 | 0.337 | **0.504** |
| F1 | 0.597 | 0.497 | 0.553 | **0.657** | 0.73 | 0.693 | 0.694 | **0.744** |
| Acc | 0.594 | 0.498 | 0.477 | **0.63** | 0.729 | 0.688 | 0.643 | **0.741** |

(a) CMU-MOSI dataset

(b) Hateful Memes dataset

Figure 7: Models' performances when being exposed to different label ratios.

Table 7: Semi-supervised learning with 50% labelled data on CMU-MOSI and Hateful Memes datasets.

| Modality | Metrics | Framework | | | | |
|---|---|---|---|---|---|---|
| | | Unimodal | Prompt-Trans | GMC | ActionMAE | Robult |
| **CMU-MOSI** | | | | | | |
| Text | MAE | 1.157 | - | 1.286 | 1.161 | **1.137** |
| | Corr | 0.502 | - | 0.384 | 0.503 | **0.516** |
| | F1 | 0.712 | - | 0.642 | 0.701 | **0.721** |
| | Acc | 0.712 | - | 0.644 | 0.709 | **0.722** |
| Audio | MAE | 1.443 | - | 1.44 | 1.603 | **1.363** |
| | Corr | 0.185 | - | 0.171 | 0.16 | **0.225** |
| | F1 | 0.563 | - | 0.556 | 0.566 | **0.569** |
| | Acc | 0.559 | - | 0.548 | 0.521 | **0.571** |
| Video | MAE | 1.514 | - | 1.458 | **1.406** | 1.429 |
| | Corr | 0.123 | - | **0.187** | 0.132 | 0.15 |
| | F1 | 0.541 | - | 0.566 | 0.551 | **0.57** |
| | Acc | 0.49 | - | 0.56 | 0.535 | **0.561** |
| Full | MAE | 1.508 | - | 1.148 | 1.096 | **1.092** |
| | Corr | 0.254 | - | 0.536 | 0.536 | **0.63** |
| | F1 | 0.566 | - | 0.709 | 0.728 | **0.761** |
| | Acc | 0.545 | - | 0.71 | 0.705 | **0.762** |
| **Hateful Memes** | | | | | | |
| Text | AUROC | 0.636 | 0.51 | 0.641 | 0.603 | **0.657** |
| | Accuracy | 0.583 | 0.508 | **0.592** | 0.556 | 0.589 |
| Image | AUROC | **0.625** | 0.528 | 0.624 | 0.592 | 0.621 |
| | Accuracy | 0.566 | 0.526 | **0.568** | 0.53 | **0.568** |
| Full | AUROC | 0.636 | 0.672 | 0.67 | 0.661 | **0.673** |
| | Accuracy | 0.57 | 0.594 | 0.596 | 0.573 | **0.604** |

Table 8: Supervised learning results on CMU-MOSI and Hateful Memes datasets.

| Modality | Metrics | Framework | | | | |
|---|---|---|---|---|---|---|
| | | Unimodal | Prompt-Trans | GMC | ActionMAE | Robult |
| **CMU-MOSI** | | | | | | |
| Text | MAE | 1.126 | - | 1.233 | 1.108 | **1.066** |
| | Corr | 0.513 | - | 0.443 | 0.498 | **0.574** |
| | F1 | 0.716 | - | 0.665 | 0.739 | **0.756** |
| | Acc | 0.717 | - | 0.667 | 0.749 | **0.753** |
| Audio | MAE | 1.421 | - | 1.414 | 1.569 | **1.392** |
| | Corr | 0.217 | - | 0.188 | 0.143 | **0.241** |
| | F1 | **0.574** | - | 0.571 | 0.569 | **0.574** |
| | Acc | 0.553 | - | **0.567** | 0.523 | 0.561 |
| Video | MAE | 1.422 | - | 1.441 | **1.532** | **1.419** |
| | Corr | 0.138 | - | **0.205** | 0.119 | 0.156 |
| | F1 | 0.535 | - | **0.552** | 0.534 | 0.513 |
| | Acc | 0.512 | - | **0.542** | 0.421 | 0.512 |
| Full | MAE | 1.191 | - | 1.093 | 1.055 | **1.011** |
| | Corr | 0.463 | - | 0.612 | 0.607 | **0.663** |
| | F1 | 0.726 | - | 0.735 | 0.757 | **0.764** |
| | Acc | 0.696 | - | 0.736 | 0.763 | **0.765** |
| **Hateful Memes** | | | | | | |
| Text | AUROC | 0.641 | 0.527 | **0.67** | 0.654 | **0.67** |
| | Accuracy | 0.585 | 0.522 | 0.607 | 0.515 | **0.63** |
| Image | AUROC | **0.639** | 0.541 | 0.64 | 0.653 | **0.665** |
| | Accuracy | 0.595 | 0.517 | **0.591** | 0.515 | 0.622 |
| Full | AUROC | 0.649 | **0.683** | 0.633 | 0.666 | 0.675 |
| | Accuracy | 0.585 | **0.634** | 0.596 | 0.536 | **0.634** |

To clearly illustrate the performance improvement of Robult and the baselines in each scenario, we provide visualizations of Pearson correlation for CMU-MOSI and AUROC for Hateful Memes in Figure 7. Among all methods, Robult demonstrates the best stability and consistency in performance, regardless of input modalities or label ratios.

## C.3 EXTENDED COMPARISON WITH RECENT FRAMEWORKS

**Baselines.** We adopt two recent approaches utilizing Constrastive Loss Xu et al. (2024) and recontruction strategy Wang et al. (2023) for more comprehensive comparision of Robult with exisiting State-of-the-art frameworks. Their original codebases are slightly adjusted for semi-supervised settings, and the dimensions of the latent space are aligned with Robult's (60) to minimize bias in the comparison.

**Settings and Result.** We evaluate the models' performance using a 5% semi-supervised task with the CMU-MOSI and CMU-MOSEI datasets, testing all possible combinations of modalities input.

Table 9: Additional comparison on CMU-MOSI and CMU-MOSEI Datasets.

| Modality | Metrics | MOSI Dataset | | | MOSEI Dataset | | |
|---|---|---|---|---|---|---|---|
| | | DiCMoR | SEM | Robult | DiCMoR | SEM | Robult |
| Text | MAE | 1.444 | 1.632 | **1.397** | 0.819 | 0.894 | **0.784** |
| | Corr | 0.085 | 0.095 | **0.144** | 0.276 | 0.252 | **0.459** |
| | F1 | 0.536 | 0.506 | **0.578** | 0.572 | 0.599 | **0.739** |
| | Acc | 0.511 | 0.53 | **0.569** | 0.657 | 0.63 | **0.732** |
| Audio | MAE | 1.504 | 1.739 | **1.415** | 0.829 | 1.037 | **0.825** |
| | Corr | 0.006 | 0.043 | **0.085** | 0.201 | 0.138 | **0.221** |
| | F1 | 0.484 | 0.441 | **0.539** | 0.537 | 0.545 | 0.679 |
| | Acc | 0.49 | 0.463 | **0.535** | 0.642 | 0.536 | **0.65** |
| Vision | MAE | 1.454 | 1.87 | **1.425** | 0.83 | 0.992 | **0.826** |
| | Corr | 0.019 | 0.017 | **0.086** | 0.163 | 0.133 | **0.201** |
| | F1 | 0.526 | 0.449 | **0.593** | 0.552 | 0.523 | **0.647** |
| | Acc | **0.524** | 0.475 | 0.522 | **0.635** | 0.524 | 0.632 |
| Text + Audio | MAE | 1.481 | 1.728 | **1.401** | 0.828 | 0.919 | **0.762** |
| | Corr | 0.013 | 0.125 | **0.141** | 0.173 | 0.248 | **0.439** |
| | F1 | 0.494 | 0.443 | **0.563** | 0.563 | 0.611 | **0.733** |
| | Acc | 0.495 | 0.465 | **0.546** | 0.639 | 0.602 | **0.717** |
| Text + Vision | MAE | 1.473 | 1.758 | **1.415** | 0.832 | 0.92 | **0.788** |
| | Corr | 0.012 | 0.077 | **0.146** | 0.158 | 0.25 | **0.399** |
| | F1 | 0.514 | 0.452 | **0.58** | 0.579 | 0.612 | **0.718** |
| | Acc | 0.514 | 0.476 | **0.534** | 0.638 | 0.592 | **0.704** |
| Audio + Vision | MAE | 1.478 | 1.794 | **1.426** | 0.836 | 0.923 | **0.812** |
| | Corr | 0.023 | 0.035 | **0.091** | 0.138 | 0.143 | **0.244** |
| | F1 | 0.484 | 0.429 | **0.581** | 0.581 | 0.535 | **0.663** |
| | Acc | 0.491 | 0.451 | **0.527** | 0.626 | 0.552 | **0.64** |
| Full | MAE | 1.468 | 1.797 | **1.392** | 0.839 | 0.902 | **0.779** |
| | Corr | 0.035 | 0.041 | **0.247** | 0.149 | 0.249 | **0.504** |
| | F1 | 0.488 | 0.432 | **0.657** | 0.587 | 0.625 | **0.744** |
| | Acc | 0.495 | 0.453 | **0.63** | 0.614 | 0.667 | **0.741** |

Table 9 summarizes the results of this study. As shown, Robult consistently outperforms the two frameworks in most scenarios. This experiment further highlights Robult's robustness in semi-supervised settings and when modalities are missing.

## C.4    Extended ablation studies on Robult design

**Setting.** In this analysis, our goal is to understand the contributions of our applied strategies to overall Robult's performance. Specifically, we adopt several ablation studies:

- **Removal of Unimodal branches** $g^i(.)(i = 1 \ldots M)$: The output of the shared branch $g^0(.)$ is directly fed into the classifier to yield the final result. The remaining framework is trained normally with the soft PU loss and downstream task loss.

- **Soft-PU Loss Ablation - Removal of weighting scheme**: Uniform weight is adopted instead of our proposed dynamic weighting scheme.

- **Soft-PU Loss Ablation - Removal of pseudo labeling**: All unlabeled samples are considered negatives, resemble normal constrastive learning scheme.

**Result.** The results, presented in Table 10, indicate an overall performance decrease across all modalities on two tested datasets. Specifically, with removal of unimodal branches, in the case of a small dataset with few labeled samples (CMU-MOSI), this ablation causes some weaker modalities to fail in generating beneficial representations during learning. Similar patterns are captures with ablations of Soft P-U loss. The results indicate a consistent decrease in performance across both variations and two test datasets. This analysis empirically supports the effectiveness of our soft PU loss.

## C.5    Extended ablation study regarding choice of RBF Kernel

In this analysis, our goal is to understand the role of our weighting scheme in the Soft P-U Loss. We compare two distinct weighting mechanisms to evaluate how closely a positive candidate matches

Table 10: Additional Ablation Study with Robult on two datasets CMU-MOSI and Hateful Memes.

| Modality | Metrics | Framework | | | |
|---|---|---|---|---|---|
| | | **Robult** | Robult w/o weighting scheme | Robult w/o unique branches | Robult w/o pseudo labelling |
| *MOSI Dataset:* | | | | | |
| Text | MAE | **1.397** | 1.418 | 1.514 | 1.412 |
| | Corr | 0.144 | 0.125 | 0.131 | **0.16** |
| | F1 | **0.578** | 0.551 | 0.53 | 0.553 |
| | Acc | **0.569** | 0.548 | 0.443 | 0.551 |
| Audio | MAE | **1.415** | 1.479 | 1.576 | 1.492 |
| | Corr | **0.085** | -0.042 | -0.096 | -0.001 |
| | F1 | **0.539** | 0.514 | 0.513 | 0.528 |
| | Acc | **0.535** | 0.456 | 0.514 | 0.455 |
| Vision | MAE | **1.425** | 1.434 | 1.509 | 1.443 |
| | Corr | 0.086 | **0.087** | 0.034 | 0.077 |
| | F1 | **0.593** | **0.593** | 0.526 | **0.593** |
| | Acc | 0.522 | 0.422 | **0.528** | 0.422 |
| Full | MAE | 1.392 | 1.388 | 1.487 | **1.359** |
| | Corr | **0.247** | 0.192 | 0.207 | 0.214 |
| | F1 | **0.657** | 0.566 | 0.567 | 0.595 |
| | Acc | **0.63** | 0.569 | 0.496 | 0.591 |
| *Hateful Memes:* | | | | | |
| Text | AUROC | **0.623** | 0.556 | 0.586 | 0.555 |
| | Accuracy | **0.59** | 0.541 | 0.577 | 0.556 |
| Image | AUROC | 0.596 | **0.597** | 0.547 | **0.597** |
| | Accuracy | **0.562** | 0.511 | 0.533 | 0.51 |
| Full | AUROC | **0.632** | 0.571 | 0.601 | 0.602 |
| | Accuracy | **0.595** | 0.345 | 0.544 | 0.51 |

the true positive pair, and then contrast these mechanisms against our initial choice of the RBF Kernel.

**Setting.** The two new weighting mechanisms are designed based on normalized distances, with the difference lying in the choice of distance measures $\delta(;)$ (here Euclidean and Manhattan distances). Specifically, within a mini-batch $B$, given the reference proximity $\phi_{ref}$ and the proximity $\phi_i$ of the positive candidate that need to be weighted, we calculated the weight as follow:

$$w_i = 1 - \tilde{d}_i;$$
$$\tilde{d}_i = \frac{\delta(\phi_i, \phi_{ref})}{\max_B \delta(\phi_j, \phi_{ref})}.$$

**Result.** We refer to the variant using a Manhattan distance-based strategy as *Robult - L1*, and the one utilizing Euclidean measures as *Robult - L2*. These two variants are evaluated against the original RBF-based model in a 5% semi-supervised task with the CMU-MOSI and CMU-MOSEI datasets. The comprehensive results are presented in Table 11. Generally, we observe minor differences in performance among the weighting schemes. While the RBF approach yields the most consistent results across various input combinations for these datasets, we do not declare it the definitive best weighting method. We believe further research is needed to identify the most appropriate strategy for the dataset of interest.

C.6 MUTUAL INFORMATION MAXIMIZATION ANALYSIS

As stated in our main text, the necessity to model the objective of learning unimodal representations to maximize mutual information with a lower bound arises because mutual information cannot be precisely calculated. This is due to the changing values of the variables over time and the discrete nature of the datasets. To verify the effectiveness of our proposed method, we adopt the histogram-based method in Peng et al. (2005) to approximate MI between two variables after the

Table 11: Additional ablation study on CMU-MOSI and CMU-MOSEI Datasets.

| Modality | Metrics | MOSI Dataset | | | MOSEI Dataset | | |
|---|---|---|---|---|---|---|---|
| | | Robult - L1 | Robult - L2 | Robult | Robult - L1 | Robult - L2 | Robult |
| Text | MAE | 1.486 | 1.456 | **1.397** | 0.793 | 0.792 | **0.784** |
| | Corr | 0.1 | **0.184** | 0.144 | 0.421 | 0.456 | **0.459** |
| | F1 | 0.571 | 0.573 | **0.578** | 0.733 | **0.741** | 0.739 |
| | Acc | 0.545 | **0.576** | 0.569 | 0.729 | **0.735** | 0.732 |
| Audio | MAE | 1.475 | 1.51 | **1.415** | **0.825** | 0.853 | **0.825** |
| | Corr | 0.049 | 0.083 | **0.085** | 0.199 | 0.165 | **0.221** |
| | F1 | **0.544** | 0.477 | 0.539 | 0.674 | 0.597 | **0.679** |
| | Acc | 0.52 | 0.478 | **0.535** | 0.635 | 0.593 | **0.65** |
| Vision | MAE | 1.475 | 1.478 | **1.425** | 0.917 | 0.931 | **0.826** |
| | Corr | 0.028 | 0.045 | **0.086** | 0.133 | 0.18 | **0.201** |
| | F1 | **0.593** | 0.582 | **0.593** | 0.567 | 0.602 | **0.647** |
| | Acc | 0.492 | **0.522** | **0.522** | 0.572 | 0.603 | **0.632** |
| Text + Audio | MAE | 1.477 | **1.395** | 1.401 | 0.764 | **0.759** | 0.762 |
| | Corr | 0.089 | **0.166** | 0.141 | 0.439 | **0.454** | 0.439 |
| | F1 | **0.57** | 0.558 | 0.563 | **0.74** | **0.74** | 0.733 |
| | Acc | 0.531 | **0.561** | 0.546 | 0.722 | **0.731** | 0.717 |
| Text + Vision | MAE | 1.47 | **1.389** | 1.415 | 0.781 | **0.772** | 0.788 |
| | Corr | 0.128 | **0.236** | 0.146 | 0.391 | **0.429** | 0.399 |
| | F1 | **0.59** | 0.553 | 0.58 | 0.705 | **0.718** | **0.718** |
| | Acc | 0.52 | **0.556** | 0.534 | 0.7 | **0.714** | 0.704 |
| Audio + Vision | MAE | 1.465 | 1.475 | **1.426** | 0.834 | 0.843 | **0.812** |
| | Corr | 0.04 | 0.054 | **0.091** | 0.187 | 0.216 | **0.244** |
| | F1 | 0.577 | 0.533 | **0.581** | 0.635 | 0.622 | **0.663** |
| | Acc | 0.472 | 0.491 | **0.527** | 0.622 | 0.618 | **0.64** |
| Full | MAE | 1.403 | **1.366** | 1.392 | 0.812 | **0.778** | 0.779 |
| | Corr | 0.223 | 0.235 | **0.247** | 0.45 | 0.438 | **0.504** |
| | F1 | 0.554 | 0.585 | **0.657** | 0.703 | 0.728 | **0.744** |
| | Acc | 0.547 | 0.583 | **0.63** | 0.708 | 0.732 | **0.741** |

Table 12: Mutual Information between fused and unimodal representations on the CMU-MOSI dataset.

| Modality | Mutual Information with fused representation | |
|---|---|---|
| | Robult | Robult w/o Soft P-U Loss |
| Text | 0.309 | 0.054 |
| Audio | 0.285 | 0.077 |
| Vision | 0.274 | 0.083 |
| Fused | **2.037** | **1.707** |

training process with and without our soft PU loss (Table 12). The result suggest two important points:

- With our soft PU loss, the mutual information of all unimodal representations with the fused representation increase significantly.

- The entropy of the fused representation also increases with the use of our loss, suggesting that the fused representation also get enriched after training with the soft PU loss.

## C.7 SOFT LABEL QUALITY ANALYSIS

We acknowledge that the quality of pseudo-labels is crucial for effective model training. This is why we incorporate our weighting scheme into the Positive-Unlabeled (PU) contrastive loss, considering the stochastic and unstable nature of pseudo-labels. This approach helps to reduce the impact of noisy pseudo-labels on the training process.

To demonstrate the effect of both pseudo-labels and our weighting strategy, we visualize the confusion matrix of pseudo-labels with and without the weighting scheme, compared to ground truth labels (Figure 8). This figure is plotted at epoch 20 of our training process using the CMU-MOSI dataset. The confusion matrix shows a strong correlation between pseudo-labels and ground truth labels, and the weighting scheme (removing all samples with weights below the $25\%$ percentile within the batch) effectively filters out some false positives identified by the pseudo labels.

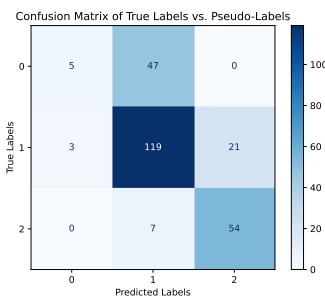
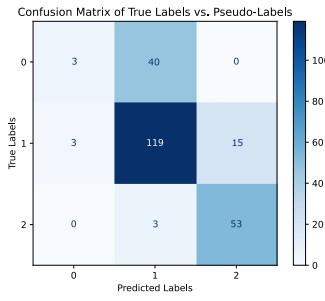

(a) Without weighting scheme

(b) With weighting scheme: Weight threshold of 25% percentile.

Figure 8: Confusion matrix of Pseudo Labels versus groundtruth label at epoch 20 on CMU-MOSI dataset.

## C.8   CLUSTERABILITY ANALYSIS

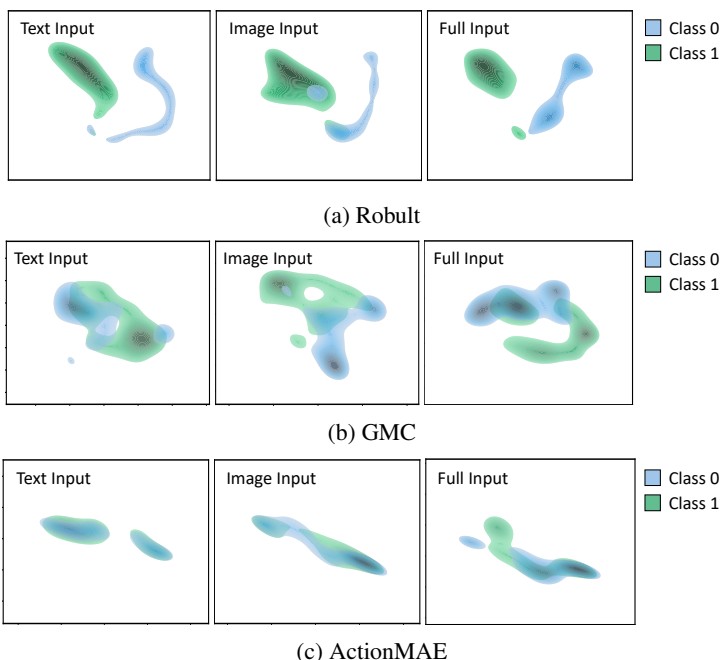

(a) Robult

(b) GMC

(c) ActionMAE

Figure 9: Representation clusters generated by different methods on Hateful Memes dataset.

Complementing the Alignment Uniformity analysis presented in the main manuscript, we provide a comparison of the clusterability characteristic of the learned representations in Figure 9. This experiment is conducted with Robult, compared to GMC and ActionMAE on Hateful Memes dataset. This qualitative analysis demonstrates that Robult's representations are better clustered even in scenarios where different modalities are missing. In contrast, the other methods do not exhibit this level of clustering effectiveness.

## C.9   TRANSFERABILITY ANALYSIS

With this experiment, we investigate the tranferability characteristic of Robult, as well as existing state-of-the-art frameworks and baselines.

**Experiment settings.**   Inspired by common pre-training procedures, where a model is initially trained on a large dataset for a source task and then fine-tuned for a target task, we designed an experiment to evaluate the zero-shot performance of all models on CMU-MOSI after being trained

Table 13: Transferability result on CMU-MOSI dataset.

| Metrics | MOSI Dataset | | | |
| | Unimodal | GMC | ActionMAE | Robult |
| --- | --- | --- | --- | --- |
| *Text Modality:* | | | | |
| MAE | 1.448 | 1.454 | 1.456 | **1.446** |
| Corr | **0.132** | 0.119 | 0.106 | **0.132** |
| F1 | 0.559 | 0.551 | **0.624** | 0.53 |
| Acc | 0.527 | 0.48 | 0.481 | **0.531** |
| *Audio Modality:* | | | | |
| MAE | 1.514 | **1.456** | 1.517 | 1.504 |
| Corr | -0.196 | **0.085** | 0.084 | -0.126 |
| F1 | 0.5 | **0.592** | 0.57 | 0.486 |
| Acc | **0.512** | 0.429 | 0.423 | 0.486 |
| *Vision Modality:* | | | | |
| MAE | 1.473 | 1.86 | 1.547 | **1.38** |
| Corr | **0.076** | -0.087 | 0.017 | 0.058 |
| F1 | 0.71 | 0.593 | 0.561 | **0.732** |
| Acc | 0.546 | 0.422 | 0.432 | **0.577** |
| *Text+Audio Modalities:* | | | | |
| MAE | 1.405 | 1.441 | **1.378** | 1.438 |
| Corr | 0.013 | **0.164** | 0.101 | 0.044 |
| F1 | 0.504 | 0.569 | **0.591** | 0.516 |
| Acc | 0.456 | 0.467 | 0.49 | **0.509** |
| *Text+Vision Modalities:* | | | | |
| MAE | 1.45 | 1.629 | 1.403 | **1.383** |
| Corr | 0.119 | 0.106 | 0.107 | **0.137** |
| F1 | **0.621** | 0.593 | 0.606 | 0.615 |
| Acc | 0.536 | 0.422 | 0.568 | **0.575** |
| *Audio+Vision Modalities:* | | | | |
| MAE | 1.434 | 1.57 | 1.528 | **1.428** |
| Corr | -0.181 | -0.005 | **0.08** | -0.088 |
| F1 | 0.496 | 0.533 | 0.551 | **0.554** |
| Acc | 0.459 | 0.422 | 0.441 | **0.487** |
| *Full Modalities:* | | | | |
| MAE | 1.456 | 1.684 | 1.472 | **1.399** |
| Corr | 0.088 | 0.072 | 0.066 | **0.202** |
| F1 | 0.549 | 0.573 | 0.55 | **0.588** |
| Acc | 0.551 | 0.425 | 0.551 | **0.585** |

with the CMU-MOSEI dataset. This setting aligns with common practices, as CMU-MOSEI is larger in scale, covering a wider range of sentiment levels and emotions compared to CMU-MOSI Zadeh et al. (2018b). To conduct the experiment, we first pretrain all methods with CMU-MOSEI using 5% labeled data, simultaneously evaluating them in a semi-supervised scenario. Since zero-shot evaluation requires no fine-tuning stage, all model architectures must remain intact after pretraining. However, there are discrepancies in the dimensions of the input data between CMU-MOSI and CMU-MOSEI. Specifically, the audio input of CMU-MOSI has a latent dimension of 5, while that of CMU-MOSEI is 74. Additionally, the video input of CMU-MOSI and CMU-MOSEI is 20 and 35, respectively. To address this issue, we generate a compact version of CMU-MOSEI by employing T-SNE on the original data, aligning the dimensions with those in the CMU-MOSI datasets. After pretraining, we directly run evaluations on the normal test set of CMU-MOSI, given different combinations of input modalities to evaluate modalities missing performance.

**Results.** Table 13 provides a summary of the results from this experiment. Generally, all methods experience a reduction in performance in certain cases when transferred to a different dataset. However, among all approaches, Robult consistently achieves the best performance, as indicated by the

recorded metrics. In addition, it is noteworthy that Robult is the only approach capable of producing meaningful results with input from full modalities in this zero-shot transfer setting.

## C.10 INCORPORATION WITH EXISTING APPROACHES

This analysis investigates the ability of Robust in incorporating with other approaches to enhance their desired characteristics in learned representations.

**Experiment settings.** We select GMC as a baseline approach for conducting this experiment. GMC aims to preserve the geometrical alignment of representations from different modalities through a geometrical contrastive loss Poklukar et al. (2022b). To observe the impact of incorporating Robult with GMC to preserve this characteristic, we simply adopt their geometrical contrastive loss with our existing $\mathcal{L}_{(u)lb}$:

$$\mathcal{L}_{lb}^i = -\frac{1}{||B_{F=1,L=1}||} \sum_{\substack{(j,k)\sim \\ p(F=1,L=1)}} \log v(s_j, z_k^i) + \log v(s_j, s_k) + \log v(z_j^i, z_k^i);$$

$$\mathcal{L}_{lb} = -\frac{1}{M} \sum_{i=1}^{M} \mathcal{L}_{lb}^i.$$

and (18)

$$\mathcal{L}_{ulb}^i = -\frac{1}{||B_{F=1,L=0}||} \sum_{\substack{(j,k)\sim \\ p(F=1,L=0)}} w_{jk}^i \left[ \log v(s_j, z_k^i) + \log v(s_j, s_k) + \log v(z_j^i, z_k^i) \right];$$

$$\mathcal{L}_{ulb} = -\frac{1}{M} \sum_{i=1}^{M} \mathcal{L}_{ulb}^i.$$

To evaluate the geometrical alignment of the learned representations, we employ Delaunay Component Analysis (DCA) Poklukar et al. (2022a), a technique similar to that used in GMC. DCA involves comparing geometric and topological properties of an evaluation set of representations (E) with a reference set (R), which acts as an approximation of the true underlying manifold. Following the evaluation strategy outlined in Poklukar et al. (2022b), we consider three metrics provided by DCA that reflect the geometric alignment between R (representations of full modalities input) and E (representations of single modality inputs): network quality $q \in [0, 1]$, precision $\mathcal{P}$, and recall $\mathcal{R}$. We report the harmonic mean defined as $3/(1/\mathcal{P} + 1/\mathcal{R} + 1/q)$ when all $\mathcal{P}, \mathcal{R}, q > 0$ and 0 otherwise. For a detailed description of DCA and its settings, please refer to the original work Poklukar et al. (2022a;b).

**Results.** We provide the alignment metrics for the representations generated with CMU-MOSI and Hateful Memes datasets, considering only $50\%$ labeled data in their respective training sets (Table 14). The statistics indicate that Robult effectively enhances the performance of GMC in its effort to preserve geometrical alignment under the constraint of limited label information. We anticipate that this behavior can potentially be extended to other methods under limited available label information, although additional investigations are needed to verify this.

Table 14: DCA Scores of models, evaluating geometrical alignment of full-modalities representations with unimodal representations.

| Dataset | R | E | Metrics | Unimodal | GMC | **Robult + GMC** |
|---------|------|--------|---------|----------|-------|------------------|
| MOSI Dataset | Full | Text | MAE | 0.473 | 0.529 | **0.535** |
| | Full | Audio | Corr | 0 | 0.375 | **0.393** |
| | Full | Vision | F1 | 0 | **0.478** | 0.335 |
| Hateful Memes | Full | Text | AUROC | 0.349 | 0.489 | **0.518** |
| | Full | Image | AUROC | 0 | 0.456 | **0.509** |

