# OpenReview forum: "Robult: A Scalable Framework for Semi-Supervised Multimodal Learning with Missing Modalities"
_ICLR.cc/2025/Conference — ICLR 2025 Conference Withdrawn Submission_

### Official Review · Reviewer_yiyF · 2024-11-02

**Soundness:** 3
**Presentation:** 3
**Contribution:** 3
**Rating:** 5
**Confidence:** 5

**Summary:**

This paper presents ROBULT, a framework designed to address modality missingness and sparse labeling issues in real-world applications, employing a two-pronged, information-theoretic approach. Specifically, to retain synergistic information across modalities, the authors propose a soft Positive-Unlabeled (PU) contrastive loss that mitigates false negatives and enhances label-level sampling. Additionally, to preserve modality-specific features, a reconstruction loss is introduced to ensure complementary information is maintained during training. Extensive experiments and analyses demonstrate ROBULT’s effectiveness across various multimodal tasks, such as sentiment analysis and multi-label classification.

**Strengths:**

1. The paper has a solid starting point, offering a practical solution for missing and semi-supervised settings in multimodal learning.
2. Detailed experiments and analysis provide strong evidence supporting ROBULT’s efficiency.

**Weaknesses:**

1. The challenges of semi-supervised learning and missing modalities have been explored before. Why does the author claim this remains an unsolved problem? (e.g., “Missing Modality Robustness in Semi-Supervised Multi-Modal Semantic Segmentation,” “Semi-Supervised Multi-Modal Learning with Incomplete Modalities”)
2. Does the training data require completeness across all modalities? If a sample has only one available modality, can it still be used in training? Excluding such data may result in lost information and underutilization of available data.
3. How did the authors simulate missing modality scenarios in their experiments?
4. In Methodology (paragraph 4), why are the inputs of $f^i, g^i, c$ written as $h^i_j, u^i_j, z^i_j$? Shouldn’t these be outputs instead?
5. The authors aim to use a classifier and the average proximity to labeled samples to identify false positives. However, why employ the RBF kernel to adjust PU loss precisely? What would happen if a true positive had low proximity to the reference? This strategy’s rationale could benefit from further clarification.
6. Could the authors provide a more detailed derivation for the transition from Equation (9) to Equation (10)?
7. The second paragraph of the introduction seems incomplete.

**Questions:**

1. The authors state in the abstract that “ROBULT seamlessly integrates into deep learning architectures.” Could they clarify how this integration works in practice?
2. How should we interpret the synergy latent variable $S$ and the unimodal variable $Z$?
3. What is the relationship between missing modalities and semi-labeled settings? In which other domains could this approach be applied?

---

### Official Review · Reviewer_Jnpu · 2024-11-02

**Soundness:** 2
**Presentation:** 3
**Contribution:** 2
**Rating:** 5
**Confidence:** 5

**Summary:**

This paper proposed a method, Robult, to address the challenges of missing modalities and limited labeled data simultaneously. Robult leveraged a latent reconstruction loss to retain unique modality-specific information and a a novel soft Positive-Unlabeled contrastive loss to efficiently utilize sparse labeled data in semi-supervised settings.

**Strengths:**

1. This paper explores the multimodal learning under the modality-missing and  semi-supervised scenarios, which is a very interesting research topic and could have a potential impact for realistic application purposes.
2. The Robult supports various modality types and quantities, making it adaptable to many existing multimodal learning frameworks.

**Weaknesses:**

1. Robult maximized the mutual information bewteen the S and the unimodal representation Z^i to capture the shared and synergy information and designed a latent reconstruction loss to retain unique modality-specific information. Actually, the idea of information decoupling has been proposed in many works, such as [1], [2], what is the difference between Robult and these works? Besides, [3] also conducted experiments under the scenarios with missing modalities and labels, how does Robult perform on their task？
2. The authors claim that designed reconstruction loss can help produce unique representations U^i for each modality, maybe some visualizations that show the difference between the distributions of U^i can better demonstrate this point.
3. How are the hyperparameters set for each loss item L_ulb, L_lb, L_rec and L_sup?  Are they all set as 1? If so, why is that? If not, how sensitive are the？

[1] Lee M, Pavlovic V. Private-shared disentangled multimodal vae for learning of hybrid latent representations[J]. arXiv preprint arXiv:2012.13024, 2020.
[2] Daunhawer I, Sutter T M, Marcinkevičs R, et al. Self-supervised disentanglement of modality-specific and shared factors improves multimodal generative models[C]//Pattern Recognition: 42nd DAGM German Conference, DAGM GCPR 2020, Tübingen, Germany, September 28–October 1, 2020, Proceedings 42. Springer International Publishing, 2021: 459-473.
[3] Wu Z, Dadu A, Tustison N, et al. Multimodal patient representation learning with missing modalities and labels[C]//The Twelfth International Conference on Learning Representations. 2024.

**Questions:**

see strength and weakness

---

### Official Review · Reviewer_2rm1 · 2024-11-04

**Soundness:** 3
**Presentation:** 4
**Contribution:** 3
**Rating:** 6
**Confidence:** 4

**Summary:**

This paper introduces Robult, a framework designed to tackle missing modalities in semi-supervised multimodal learning. Robult leverages information-theoretic methods to retain modality-specific features and enhance cross-modal synergy. Key components include a latent reconstruction loss and a soft contrastive loss for better utilization of sparse labeled data. Experimental results show Robult’s superior performance over existing methods in handling missing modalities, with a lightweight design that supports scalability and easy integration into deep learning architectures.

**Strengths:**

(1). Robult's unique design effectively improves model robustness against missing modalities through the preservation of both modality-specific and cross-modal information.

(2). Robult demonstrates superior performance over existing methods in both semi-supervised learning and missing modality scenarios. Comprehensive qualitative analyses, including representation alignment, uniformity assessments, and ablation studies, further clarify Robult’s mechanisms.

(3). The paper clearly details Robult's architecture, training process, and experimental setup, and includes comprehensive code and data preparation instructions, enabling replication and enhancing transparency.

(4). Robult's design allows for seamless integration with current deep learning frameworks, highlighting its practical value for real-world applications with missing modality issues.

**Weaknesses:**

Insufficient Experiments: More experiments should be provided, including sensitivity analysis of parameters and exploration of various multi-modal fusion strategies.

**Questions:**

(1). Could you provide more insights into the choice of the hyperparameter $\tau$? Specifically, how does $\tau$ influence model performance? Relevant experimental results should be included.

(2). As a method for multi-modal fusion strategy plugins, could you provide insights into the performance of methods under other fusion strategies or architecture?

---

### Official Review · Reviewer_iGkn · 2024-11-04

**Soundness:** 3
**Presentation:** 3
**Contribution:** 2
**Rating:** 3
**Confidence:** 4

**Summary:**

This paper introduces ROBULT, a framework designed to tackle the challenges posed by missing modalities and limited labeled data in multimodal learning. ROBULT employs an information-theoretic approach and introduces two key objectives: (1) a latent reconstruction loss to preserve unique modality-specific information, and (2) a soft Positive-Unlabeled (PU) contrastive loss to effectively leverage sparse labeled data in semi-supervised settings. ROBULT enhances performance across multiple downstream tasks while ensuring robustness even when modalities are absent during inference. Empirical results from diverse datasets demonstrate that ROBULT outperforms existing methods in managing both semi-supervised learning and missing modalities.

**Strengths:**

1.	ROBULT successfully maintains synergy while explicitly preserving the unique information of each modality, even when labels are missing during training and modalities are absent during testing. This is supported by experimental results and ablation studies.

2.	The supplementary materials provide sufficient implementation details, enabling the reproduction of the experimental results.

3.	Experimental results demonstrate the superiority of ROBULT over multiple existing baseline methods.

4.	The theoretical proof accompanying the ROBULT method establishes a solid foundation for its application.

**Weaknesses:**

1. The research question in this paper is incremental. While the study addresses the challenge of multimodal data with missing modalities and missing labels during training, it is essential to emphasize why resolving this issue is important in practical deep learning. What severe consequences could arise if this problem remains unaddressed? Including a discussion on these aspects would significantly enhance the perceived importance of the research.

2. The author claims that ROBULT is designed to tackle the challenge of multimodal data with missing modalities, and labels during training. However, the scope of the paper is quite narrow. It focuses only on the scenario where training samples include all modalities, with some samples unlabeled and certain modalities missing in the testing data. Other scenarios, such as missing modalities in the training data or varying ratios of missing modalities [R1], should also be considered to offer a more comprehensive analysis.

3. The semi-supervised experimental setup lacks comprehensiveness. The paper only examines 5% and 50% labeled training data, but in semi-supervised learning, existing works [R2, R3] typically explore a wider range of labeled training data settings, including 5%, 10%, 20%, and 50% labeled samples.

4. The writing in this paper requires improvement. The second paragraph of the "Introduction" chapter appears unfinished and would benefit from further development.

[R1] Ma M, Ren J, Zhao L, et al. Smil: Multimodal learning with severely missing modality[C]//Proceedings of the AAAI Conference on Artificial Intelligence. 2021, 35(3): 2302-2310.

[R2] Zou Y, Zhang Z, Zhang H, et al. PseudoSeg: Designing Pseudo Labels for Semantic Segmentation[C]//International Conference on Learning Representations. 2021.

[R3] Zou Y, Choi J, Wang Q, et al. Learning representational invariances for data-efficient action recognition[J]. Computer Vision and Image Understanding, 2023, 227: 103597.

**Questions:**

Regarding the soft Positive-Unlabeled (PU) contrastive loss, since this method is designed to address the issue of missing training labels, could the authors further analyze its effectiveness or contribution within ROBULT from the perspective of the quantity of labeled training data? For instance, while the authors have provided experiments with 5% and 50% labeled training data, could additional experiments or discussions be included to explore the possibility that the fewer labeled training samples there are, the greater the contribution of this proposed contrastive loss?

---

### Note · Authors · 2024-11-13

I have read and agree with the venue's withdrawal policy on behalf of myself and my co-authors.